# A Framework for Improving Wall-to-Wall Canopy Height Mapping by Integrating GEDI LiDAR

Cangjiao Wang [1,2], Andrew J. Elmore [2], Izaya Numata [2,3], Mark A. Cochrane [2], Shaogang Lei [1,*], Christopher R. Hakkenberg [4], Yuanyuan Li [1], Yibo Zhao [1] and Yu Tian [1]

1. Engineering Research Center of Ministry of Education for Mine Ecological Restoration, China University of Mining and Technology, Xuzhou 221116, China; wangcangjiao@cumt.edu.cn (C.W.); yyli@cumt.edu.cn (Y.L.); zhaoyibo@cumt.edu.cn (Y.Z.); tiany@cumt.edu.cn (Y.T.)
2. Appalachian Laboratory, University of Maryland Center for Environmental Science, 301 Braddock Rd., Frostburg, MD 21532, USA; aelmore@umces.edu (A.J.E.); izaya.numata@umces.edu (I.N.); mark.cochrane@umces.edu (M.A.C.)
3. Geospatial Sciences Center of Excellence, South Dakota State University, 1021 Medary Ave., Wecota Hall Box 506B, Brookings, SD 57007, USA
4. School of Informatics, Computing, and Cyber Systems, Northern Arizona University, 1295 Knoles Drive, Flagstaff, AZ 86011, USA; chris.hakkenberg@nau.edu
* Correspondence: lsgang@126.com

**Abstract:** Spatially continuous canopy height is a vital input for modeling forest structures and functioning. The global ecosystem dynamics investigation (GEDI) waveform can penetrate a canopy to precisely find the ground and measure canopy height, but it is spatially discontinuous over the earth's surface. A common method to achieve wall-to-wall canopy height mapping is to integrate a set of field-measured canopy heights and spectral bands from optical and/or microwave remote sensing data as ancillary information. However, due partly to the saturation of spectral reflectance to canopy height, the product of this method may misrepresent canopy height. As a result, neither GEDI footprints nor interpolated maps using the common method can accurately produce spatially continuous canopy height maps alone. To address this issue, this study proposes a framework of point-surface fusion for canopy height mapping (FPSF-CH) that uses GEDI data to calibrate the initial wall-to-wall canopy height map derived from a sub-model of FPSF-CH. The effectiveness of the proposed FPSF-CH was validated by comparison to canopy heights derived from (1) a high-resolution canopy height model derived from airborne discrete point cloud lidar across three test sites, (2) a global canopy height product (GDAL RH95), and (3) the results of the FPSF-CH sub-model without fusing with the GEDI canopy height. The results showed that the RMSE and rRMSE of FPSF-CH were 3.82, 4.05, and 3.48 m, and 18.77, 16.24, and 13.81% across the three test sites, respectively. The FPSF-CH achieved improvement over GDAL RH95, with reductions in RMSE values of 1.28, 2.25, and 2.23 m, and reductions in rRMSE values of 6.29, 9.01, and 8.90% across the three test sites, respectively. Additionally, the better performance of the FPSF-CH compared with its sub-model further confirmed the effectiveness of integrating GEDI data for calibrating wall-to-wall canopy height mapping. The proposed FPSF-CH integrates GEDI LiDAR data to provide a new avenue for accurate wall-to-wall canopy height mapping critical to applications, such as estimations of biomass, biodiversity, and carbon stocks.

**Keywords:** canopy height; FPSF-CH; calibration; GEDI LiDAR; forest

## 1. Introduction

Forest canopy height is a critical parameter for the estimation and monitoring of above-ground biomass (AGB), terrestrial carbon stocks, biodiversity, and wildlife habitat [1–5]. Given a strong relationship between canopy height and AGB, canopy height is also an essential variable for initiating forest carbon models in climate mitigation planning [6].

Moreover, canopy height has become a top influential factor for the diversity evaluation of flora and fauna [7]. Therefore, precisely measuring canopy height is fundamental for estimations of forest structure and functioning.

In the last three decades, light detection and ranging (LiDAR) has become a promising way to measure canopy height by actively transmitting laser pulses that penetrate a canopy and then reflect interaction energy from surfaces. With this penetration ability, LiDAR plays an unparalleled role in canopy height measurements compared to optical and synthetic aperture radar (SAR) datasets [8,9]. At present, several LiDAR platforms have been used for canopy height measurements, including terrestrial (including static and mobile platforms), airborne (including manned and unmanned aerial vehicles), and spaceborne LiDAR. Airborne and terrestrial LiDARs have high resolution, and are accurate in canopy height measurements, but they are limited by coverage and are expensive and time-consuming in data collection [10–12]. Meanwhile, the accuracy of canopy height measurements using different sources of airborne LiDAR is influenced by point density, forest stand density, and tree size [13]. As a result, canopy height measurements over a large-scale extent are a challenging endeavor. Fortunately, current and upcoming spaceborne LiDAR systems meet the increasing demand for accurate regional or even global canopy height measurements [14,15]. The global ecosystem dynamics investigation (GEDI), launched in December 2018 [16] provides near-global coverage observations ranging from 51.6°S to 51.6°N up until at least September 2023 (http://database.eohandbook.com/database/instrumenttable.aspx) [17]. GEDI also possesses an accuracy of less than 1 m bias in canopy height measurement [16,18]. It should be noted that GEDI footprints are not continuous across space due to discrete sampling along and across orbit tracks. As a result, using GEDI data alone cannot directly provide users with wall-to-wall canopy height maps.

To achieve wall-to-wall canopy height mapping, a common method is to integrate a set of field-measured canopy heights with optical and/or microwave remote sensing imagery as ancillary information [14,19–21]. Herein, linear regression, random forest (RF), support vector machine, etc., are commonly used to build the mapping between canopy height values and remote sensing imagery [22]. The imagery used in a common method is generally derived from the Moderate Resolution Imaging Spectroradiometer (MODIS), Landsat, Sentinel-1 (C-band SAR), and Sentinel-2 (Optical) along with other complementary data such as thematic maps of elevation, topographic slope, land cover, etc. [17,21,23,24]. These images can reveal the physiological and biochemical parameters that are potentially related to the structure and function of vegetation [25]. In this case, through integrating ancillary imagery, wall-to-wall canopy height maps can be produced across regions, nations, and even at near-global extents [14,19,20]. However, some limitations should be emphasized when integrating ancillary imagery for wall-to-wall canopy height mapping. Firstly, universal direct relationships between canopy height and imagery are confounded by many factors, including shadow, density, and forest type [26,27], which result in site-specific relationships. A site-specific relationship then hinders large-scale wall-to-wall canopy height mapping [28]. For example, disparate relationships between Landsat spectral images and canopy heights have been reported in different studies, such as a negative relationship between spectral reflectance of the near-infrared and canopy height in [28], versus flat in [29,30], and positive relationships in [31]. The second limitation is spectral saturation to canopy height, which can lead to an underestimation of the canopy height in tall or dense forests. Various examples of saturation in tall canopy heights have been reported when integrating ancillary imagery for wall-to-wall canopy height mappings, such as 20 m in [32], 25 m in [33], 27 m in [34], and 30 m in [19]. Finally, algorithms of common methods may introduce errors in canopy height mapping. For example, machine learning algorithms, employed to build the relationship between field-measured canopy heights and ancillary remote sensing imagery, may be unable to predict canopy heights outside of the range in the trained model [17,20].

As mentioned above, using GEDI data or the common method alone is problematic for an accurate wall-to-wall canopy height mapping, but the two may be complementary

when used together. Therefore, this paper proposes a framework for point-surface fusion for spatially continuous canopy height mapping (FPSF-CH). The FPSF-CH firstly used a sub-model (e.g., random forest model) as the common method to generate an antecedent wall-to-wall canopy height map by integrating field-measured canopy heights and wall-to-wall imagery derived from Sentinel-1, Sentinel-2, and along with other thematic maps. Then, the antecedent wall-to-wall canopy height map was improved by integrating GEDI observations based on a weighted linear regression. The effectiveness of the FPSF-CH was assessed through accuracy comparisons with (1) GEDI LiDAR in two study areas (1 × 1° tile region) in northeast America; (2) three high-resolution canopy height maps derived from airborne discrete point cloud LiDAR at an observation platform in National Ecological Observatory Network (NEON) sites within the two study areas; and (3) the global canopy height product, global land analysis, and discovery (GDAL), generated by Potapov et al. (2019) (hereafter referred to as GDAL RH95).

## 2. Materials and Methods

To generate an antecedent wall-to-wall canopy height map, we sought to combine a set of field-measured canopy height values with ancillary imagery in the FPSF-CH (step 1 in Figure 1), which is then processed with a calibration procedure (from step 2 to step 4). Specifically, the general RF model was employed as a sub-model in the PSFM-CH to generate an antecedent canopy height map (RF-CH) using Sentinel-1 (radar), Sentinel-2 (optical), along with other thematic maps as predictors (See Section 2.3). For calibration, we screened good-quality GEDI footprints within a focal area with forest maturity level indicated by the *k*-mean classification of the disturbance index (DI). The GEDI data screening is expected to increase the representativeness of GEDI observations of the RF-CH pixel (step 2 in Figure 1). Then, relative to the focal location, the DI difference between GEDI observations and the corresponding RF-CH pixel, and waveform sensitivity are calculated to estimate the weight for each GEDI observation (step 3 in Figure 1). The relationship between the RF-CH and the canopy height of good-quality GEDI observations is then built using weighted linear regression, which results in a final corrected canopy height (C-CH) (step 4 in Figure 1). Finally, the performance of C-CH is validated as shown in step (5) in Figure 1.

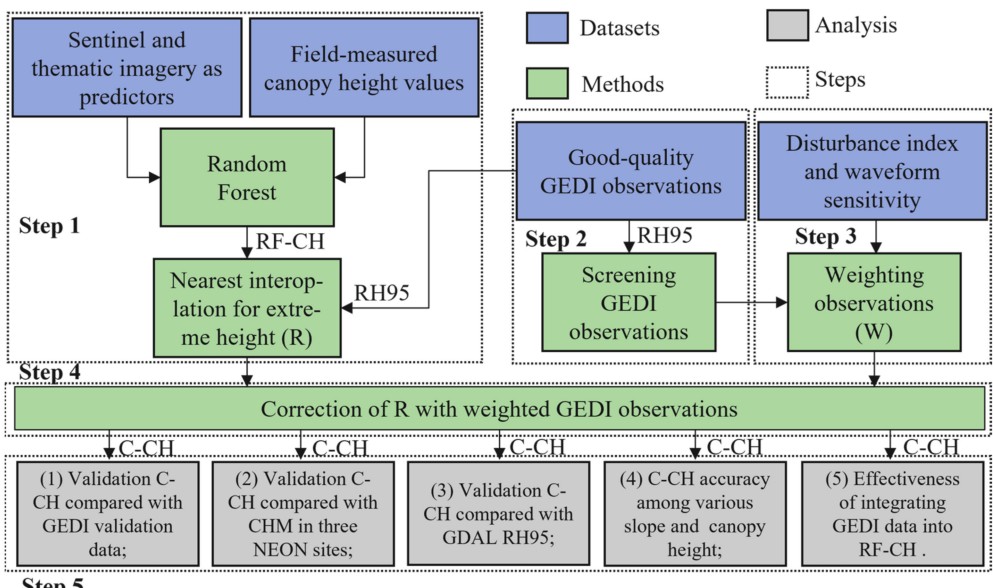

**Figure 1.** Flowchart of the FPSF-CH and its performance analysis. The RF-CH, RH95, and C-CH are canopy heights derived from RF, GEDI waveform, and the FPSF-CH. R and W are the nearest interpolation dataset and weight for each GEDI data point, respectively. CHM is the canopy height model.

## 2.1. Coupling Field-Measured Canopy Height Values with Wall-To-Wall Imagery Using RF

For the wall-to-wall RF-CH mapping, a series of field-measured canopy height values and ancillary imagery extracted from Sentinel-1, Sentinel-2, and other datasets were used to train the RF model. The RF-CH with a canopy height greater than thirty meters was replaced with GEDI data using the nearest interpolation.

## 2.2. Calibration RF-CH by Integrating GEDI LiDAR

The calibration assumes that RF-CH can distinguish short from medium-height vegetation, but cannot accurately represent very tall canopy height due to the insensitivity of imagery predictors to canopy height. Conversely, GEDI data is expected to accurately estimate tall canopy height due to its penetration ability. Hence, it is reasonable that integrating GEDI data could improve RF-CH. Assuming a linear relationship between RF-CH and the canopy height of GEDI LiDAR, the correction of RF-CH can be expressed as Equation (1).

$$\mathbf{C} = \boldsymbol{\beta} \times \mathbf{R} + \boldsymbol{\varepsilon} \tag{1}$$

where $\mathbf{C}$ is the corrected canopy height (C-CH), $\mathbf{R}$ is RF-CH, $\boldsymbol{\beta}$ is a coefficient matrix, and $\boldsymbol{\varepsilon}$ represents model residuals.

It should be noted that misusing GEDI observations for RF-CH calibration may result in unnecessary errors. For example, GEDI observations of grassland will not robustly contribute to forest canopy height calibrations. Additionally, the accuracy of GEDI waveforms for estimating canopy height is dependent on the degree of light penetration through a canopy (defined as waveform sensitivity) [33,35,36]. To avoid these errors, it is vital to search for representative GEDI observations for calibrating the corresponding RF-CH pixel (hereafter referred to as an object pixel in the RF-CH).

### 2.2.1. Searching for Corresponding GEDI Observations for Calibration

GEDI waveforms were selected for each object pixel in the RF-CH using a moving window approach. Within a defined focal distance of 6 km (equivalent to 200 pixels at a 30 m resolution), GEDI observations (constrained to be the same land cover type as the focal location) were selected for calibrating an object pixel in the RF-CH. This process ensured that canopy height was corrected using GEDI waveforms from similar land cover types. Rather than using a separate land cover dataset, we used a *k*-means clustering approach [37] to generate data clusters associated with each focal pixel. Given the positive correlation between canopy height and stand age, a *k*-means clustering classifier needs predictors that can distinguish stand ages across forest stands. For this, we used the disturbance index (DI = brightness − greenness + wetness), which was generated from the tasseled cap space components of brightness, greenness, and wetness [38]. Technically, DI represents the distance in tasseled cap space between a pixel and the area that is simultaneously the brightest and furthest from greenness and wetness extremes. Due to its ability to separate dense from sparse forest covers, the DI is an appealing index for the estimations of forest stand age and forest structure [39,40], and, thus, is used as the input for the *k*-means classifier in this paper.

Given that ancillary imagery is an accurate predictor for short vegetation mapping, but less accurate for high canopies, calibration should give more attention to tall rather than short canopies [19,32]. Focusing on tall canopy height calibration, GEDI observations were selected possessing a canopy height ($CH$) greater than the mean RF-CH ($H_{mean}$). To ensure a separatable top component of vegetation from a ground signal, waveforms were retained when their waveform mode was greater than 2 (*mode* > 2) (Equation (2)).

$$\text{Con} = \begin{cases} 1; \ \text{type}_{\text{GEDI}} = \text{type}_{\text{object}}, \ \text{CH} > \text{H}_{\text{mean}} \ \text{mode} > 2 \\ 0; \ \text{otherwise} \end{cases} \tag{2}$$

where *Con* equal to 1 represents employed, otherwise unemployed, a GEDI observation for calibration. The $type_{GEDI}$ and $type_{object}$ are the land cover types generated from the *k*-mean classifier of GEDI observations and the object pixel of RF-CH, respectively.

Doubtless, GEDI observations are unevenly distributed over space due to the influence of topography and laser transmission conditions (e.g., cloudy and rainy), which may lead to unbalanced GEDI observations for RF-CH calibration. Moreover, too many observations increase compute time, while too few observations impact total representativeness, and may introduce noise into the calibration process. To avoid the impacts of uneven GEDI observations over space on the calibration, ten GEDI observations with *Con* equal to 1 were selected for calibration with the lowest DI distances between GEDI and object pixel in the RF-CH selected.

### 2.2.2. Weighting Each GEDI Observation

Ideally, representative GEDI observations can improve estimations of the canopy height for an object pixel. Extracting exactly representative GEDI data is not feasible due partly to the differences between the set of corresponding GEDI data and object pixels. All else being equal, the less the spectral distance between the location of GEDI observations and the corresponding object pixels in the RF-CH, the more similar the observations are to the object pixel in the RF-CH. For large-size GEDI footprints, the higher the waveform sensitivity, the lower the errors in the GEDI canopy height estimation are [18]. Thus, the weight of spectral distance is represented by the DI difference ($\mathbf{W_{os}}$) and sensitivity ($\mathbf{W_{sen}}$) calculated to represent the effects of factors on the correlation of canopy height between an object pixel in the RF-CH and each corresponding GEDI observation (Equation (3)).

$$\mathbf{W}(x_i, y_i) = \mathbf{W_{os}}(x_i, y_i) \ * \ \mathbf{W_{sen}}(x_i, y_i) \tag{3}$$

where $\mathbf{W}$ is the weight for each corresponding observation. $(x_i, y_i)$ are the coordinates of the *i*-th GEDI observation. $*$ denotes element-wise product.

To account for the effects of geolocation uncertainty of GEDI observations on canopy height estimation, a non-local filter [41], using the DI difference between an object pixel in the RF-CH and the corresponding GEDI observations within a $3 \times 3$ pixel focal area, was used to weight the $\mathbf{W_{os}}$. Thus, the $\mathbf{W_{os}}$ can be quantified by Equation (4).

$$f(x_i, y_i) = 1 \ / \ T(x, y) \exp\left( - \ \mathbf{G} \ * \ \|\mathbf{S}(x_i, y_i) - \ \mathbf{S}(x_{n/2+1}, y_{n/2+1})\|_2^2 \ / \ h^2 \right) \tag{4}$$

where *G* is a Gaussian kernel weighting of the focal area pixels. S represents the factor block (e.g., DI) of a focal area. *j* is the *j*-th corresponding GEDI observation. $\|\cdot\|_2^2$ is the Euclidean distance. $h^2$ (set as 0.15) represents the noise level of the features. $T(x, y)$ is the total weight of all corresponding GEDI observations for each object pixel in the RF-CH, that can normalize the weight value so that the numerical range of $f()$ is (0, 1). Moreover, the sensitivity was normalized into (0, 1) based on min-max scaling with maximum and minimum sensitivity values of 0.9 and 1.0, respectively.

### 2.2.3. Calibrating RF-CH by Integrating Weighted GEDI Observations

By considering the effect factors for each corresponding GEDI observation, the regression coefficient parameter $\boldsymbol{\beta}$ in Equation (1) was evaluated as follows:

$$\boldsymbol{\beta} = (\mathbf{R}^T * \mathbf{W} * \mathbf{R})^{-1}\mathbf{R}^T * \mathbf{W} * \mathbf{C} \tag{5}$$

Knowing the coefficient $\boldsymbol{\beta}$, the RF-CH can be corrected by the GEDI observations. Focusing on tall canopy height calibration, RF-CH greater than $H_{mean}$ was corrected using the following:

$$\mathbf{C} = (1 - \gamma) * \mathbf{R} + \gamma * \boldsymbol{\beta} * \mathbf{R} \tag{6}$$

where $\gamma$ is the coefficient to adjust the correct extent according to RF-CH. Since RF-CH errors increased with canopy height, $\gamma$ was further expressed as follows,

$$\gamma = \begin{cases} 1; & \text{if } (\mathbf{R} - \mathrm{H}_{mean}) / (\mathrm{H}_{max} - \mathrm{H}_{mean}) > 1 \\ 0; & \text{if } \mathbf{R} < \mathrm{H}_{mean} \\ (\mathbf{R} - \mathrm{H}_{mean}) / (\mathrm{H}_{max} - \mathrm{H}_{mean}); & \text{otherwise} \end{cases} \tag{7}$$

where $H_{max}$ is the maximum canopy height of GEDI observations represented by the 95th quantiles of GEDI canopy height values. Finally, a $3 \times 3$ focal-area median filter was applied to the calibrated canopy height map which preserved larger-scale patterns while removing fine grain noise.

### 2.3. Experimentals and Analysis

#### 2.3.1. Study Area

Two experimental areas including Area 1 ranging from $-71.70°$ to $-70.70°$W, and $43.50°$ to $44.50°$N, and Area 2 ranging from $-72.60°$ to $-71.60°$W, and $42°$ to $43°$N, were selected to examine and validate the FPSF-CH. These two areas are located in one of the most densely forested northeast ecoregions of the conterminous USA, with a mean forest density of 0.67 km$^2$/km$^2$ [42]. Forest types in these two areas include deciduous, evergreen, mixed forest, and woody wetlands, in which various tree species are found such as American beech (*Fagus grandifolia*), eastern hemlock (*Tsuga canadensis*), red maple (*Acer rubrum*), and northern red oak (*Quercus rubra*). Over half of the forested areas were deforested and have since undergone regeneration, resulting in a high diversity of canopy height with different stand ages. The two study sites were distinguished from each other by their different topographic average slopes of 8.48° in Area 1 and 13.65° in Area 2, and different amounts of average tree cover, 82.00% in Area 1 and 79.00% in Area 2. These two study areas contain three NEON sites, including the Bartlett Experimental Forest NEON (BART), Harvard Forest & Quabbin Watershed NEON (HARV), and Lower Hop Brook NEON (HOPB) (Figure 2). Details of these three sites are listed in Table 1.

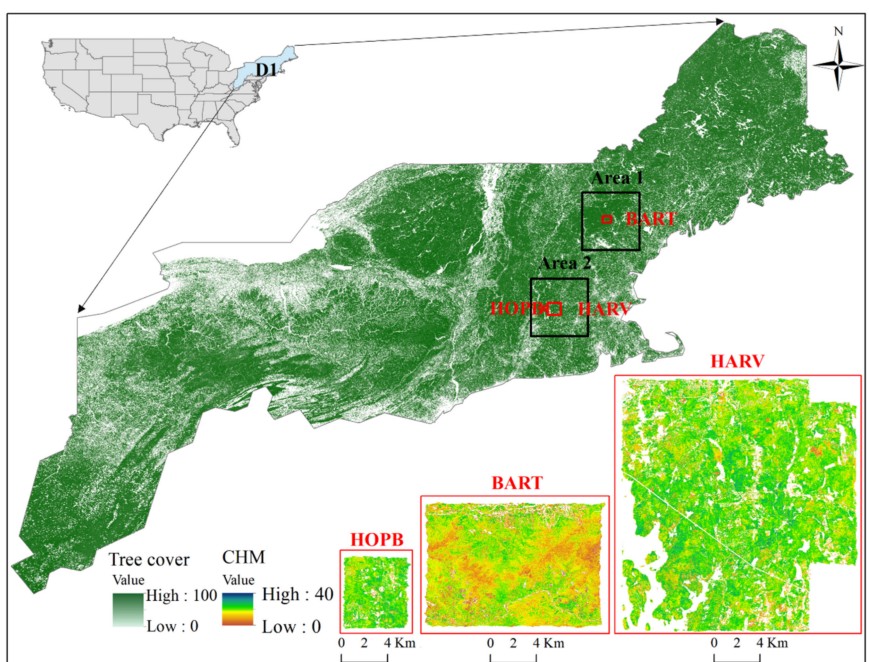

**Figure 2.** Locations of study areas. The black rectangles show the locations of Area 1 and Area 2 with $1° \times 1°$ tiles in the ecoregion domain of the North American eastern forests. Three red rectangles represent the locations of three NEON sites, including the BART, HOPB, and HARV. The tree cover

map was downloaded from the Global Land Analysis and Discovery website (https://glad.umd.edu/dataset, accessed on 27 April 2020). The canopy height models (CHMs) in BART, HOPB, and HARV are at the same scale with data ranging from 0 to 40 m.

**Table 1.** Geographic characteristics of the BART, HOPB, and HARV sites. The abbreviations of MAT, MAP, and MCH refer to mean annual temperature, mean annual precipitation, and mean canopy height, respectively.

| Sites | Latitude (°) | Longitude (°) | Mean Elevation (m) | MAT (°C) | MAP (mm) | MCH (m) |
|-------|-------------|--------------|--------------------|----------|----------|---------|
| BART  | 44.0639     | −71.2874     | 274                | 6.20     | 1325     | 23      |
| HARV  | 42.5369     | −72.1727     | 348                | 7.40     | 1199     | 26      |
| HOPB  | 42.4719     | −72.3295     | 203                | 7.90     | 1368     | 26      |

2.3.2. NEON High-Resolution CHM and DTM Datasets and Their Preprocessing

The one-meter resolution canopy height model (CHM) and digital terrain model (DTM) were downloaded from the NEON website for the BART, HARV, and HOPB sites [43,44]. Both the CHM and DTM datasets were generated from the NEON-released airborne observation platform (AOP) discrete point cloud LiDAR datasets (hereafter referred to as NEON LiDAR). The NEON LiDAR in BART, HARV, and HOPB sites were all collected during the peak greenness of the growing season in 2019, using the Optech ALTM Gemini instrument. Details of the instrument can be found in Appendix A Table A1. The CHM derived from NEON LiDAR possessed RMSE values lower than 1 m (NEON.DOC.002 387), while the DTM bias was lower than 0.15 m in flat open areas. To be comparable to the 30 m canopy height map, the 95th quantiles of the 1 m resolution CHM within each 30 m pixel (hereafter referred to as NEON RH95) were calculated as reference. The utility of NEON RH95, rather than the top of canopy height, is that the RH95 has a maximum correlation with canopy height derived from the GEDI data as shown in previous studies [18,19]. The topographic slope was calculated from the NEON DTM. Then, the median value within a 30 m resolution pixel was employed to represent the average topographic slope.

2.3.3. GDAL RH95 and Pre-Processing

The GDAL 30 m resolution global RH95 canopy height map was produced in 2019 [19]. The RH95 is defined as the elevation difference between ground elevation and the elevation where the accumulated waveform energy is 95% of the total. Given the same prediction, canopy height (RH95), spatial resolution, and concurrent year as the results in this paper, the GDAL RH95 is a good counterpart for validating the FPSF-CH. Thus, the GDAL RH95 datasets across the two study areas were downloaded from Global Land Analysis and Discovery (URL: https://glad.umd.edu/dataset/gedi/, accessed on 1 April 2022) website. Water, snow/ice, and no data areas were masked out from the GDAL RH95.

2.3.4. Tree Canopy Cover Map and Pre-Processing

To analyze the performance of the FPSF-CH among different canopy covers, we employed a 30 m resolution canopy cover map produced by the U.S. Forest Service (USFS) Geospatial Technology and Applications Center (GTAC) in 2016. The tree canopy cover map was downloaded from the U.S. Forest Service (ULR: https://data.fs.usda.gov/geodata/rastergateway/treecanopy-cover/, accessed on 27 April 2020) website.

2.3.5. GEDI Datasets and Their Pre-Processing

The second version (V2) of GEDI level-2A (L2A) datasets from June to September of 2019–2021 was downloaded from NASA LP DAAC (The Land Processes Distributed Active Archive Center: https://lpdaac.usgs.gov/product_search/). To avoid the high noise levels at the top of the canopy, RH95 was used to represent canopy height in this paper. Good-quality GEDI data were recognized as footprints in leaf-on season (leaf_off_flag equal to 0) with quality_flag equal to 1, degrade_flag equal to 0, landsat_water_persistence

lower than 10%, urban_proportion equivalent to 0, and topographic slope lower than 25°. The utility of 25° of the topographic slope is considered conservative for canopy height measurements of the GEDI waveform [35,45,46]. Notably, GEDI used six waveform preprocessing algorithms that used different thresholds for detecting signals above the noise and smoothing waveforms. The optimum algorithm (aN) for waveform pre-processing recommended by the GEDI V2 products was used in this paper.

For canopy height mapping, it is hard to collect representative field-measured datasets for model training and validation over a large-scale region. An alternative way is to employ high-quality GEDI data instead. The high-quality GEDI data were proposed to be theoretically precise for canopy height measurements, and were selected by using stricter screening conditions rather than the above filtered good-quality GEDI observations. Compared to conditions for good-quality filtering, the stricter conditions for high-quality GEDI data filtering should also meet the following criteria: being sampled by power beams rather than coverage beams, being collected with a sensitivity greater than 0.98, and possessing a slope lower than 15 degrees. As a result, GEDI data was split into three individual parts, including (1) high-quality training data, (2) validation data for training the RF model and validating canopy height mapping results, and (3) good-quality data not containing high-quality GEDI data for calibrating the RF-CH (Table 2).

**Table 2.** The descriptive statistics of training, validation, and good-quality GEDI waveforms. The N is the number of observations with a mean and standard deviation (SD) of canopy height.

| Study Area | Year | Training Data | | Validation Data | | Good-Quality Data | |
|---|---|---|---|---|---|---|---|
| | | N | Mean ± SD (m) | N | Mean ± SD (m) | N | Mean ± SD (m) |
| Area 1 | 2019 | 2220 | 21.09 ± 6.64 | 2220 | 20.75 ± 5.59 | 52,425 | 17.86 ± 6.89 |
| | 2020 | 4843 | 20.06 ± 11.34 | 4843 | 19.82 ± 11.54 | 65,853 | 17.26 ± 8.77 |
| | 2021 | 93 | 17.75 ± 4.88 | 93 | 16.59 ± 5.49 | 26,517 | 17.93 ± 7.50 |
| Area 2 | 2019 | 2130 | 23.22 ± 5.80 | 2130 | 23.41 ± 5.97 | 30,936 | 20.31 ± 7.99 |
| | 2020 | 3882 | 23.06 ± 5.47 | 3882 | 23.09 ± 5.45 | 71,539 | 20.24 ± 7.95 |
| | 2021 | 2235 | 23.21 ± 5.35 | 2235 | 23.17 ± 4.99 | 41,769 | 20.26 ± 7.82 |

### 2.3.6. Predictors for Training a RF Model

A total of 49 features used as predictors in the RF model (Appendix A Table A2) were resampled to a 30 m resolution. The RF model was trained by setting training parameters for 600 decision trees, a maximum tree depth of 10, 100 as the minimum number of samples to split a node, and 20 as the minimum number of samples in a leaf. Detailed model parameters, including the influence of each parameter, can be found in [20]. To extract the features for RF training, all available Sentinel-2 level-2A imagery ranging from June to September (growing season) in 2019 was collected. The cloud, snow, and shadow on these images were masked out according to MSK_CLDPRB (cloud probability) lower than 5, MSK_SNWPRB (cloud probability) lower than 5, and SCL (scene classification map) never equal 0. Waterbodies and developed areas were masked using the National Land Cover Database in 2019 (NLCD 2019). Then, the annual composite spectral bands were developed using the ee.ImageCollection.median() function on the Google Earth Engine (GEE) platform for spectral indices calculation (Appendix A Table A2). In addition, the Sentinel-1 SAR backscatter ARD data were composited on GEE, in which two polarization modes ('VH' or 'VV') in 2019 were used. The VV (or VH) is the polarized mode of vertical send and vertical (or horizontal) receive. These Sentinel-1 datasets were processed for additional border noise removal, speckle filtering, and radiometric terrain normalization. Then, statistical features of VV, VH, and normalized backscatter of VV and VH in the growing season and non-growing season (not from June to September) were individually extracted from the processed Sentinel-1 imagery. Due to the strong relationship between canopy height and forest stand age [39], the time since last change (SCLAST) [47], representing forest regrowth after disturbance, was used as a feature. The

SCLAST was extracted from the land change monitoring, assessment, and projection (LCMAP) products of the U.S. Geological Survey (USGS) (https://www.usgs.gov/special-topics/lcmap, accessed on 15 November 2020). Considering topographic elevation and slope were key factors for modeling canopy height, these two parameters were extracted from the shuttle radar topography mission (SRTM) [48] and counted as features.

*2.4. Effectiveness Analysis of the FPSF-CH*

To validate the effectiveness of the FPSF-CH, the accuracy of the C-CH was compared with the GEDI high-quality observations, the NEON RH95 and the GDAL RH95, taking the Pearson coefficient correlation (r), bias, mean absolute errors (MAE), root mean square error (RMSE), relative bias (rBias), and relative RMSE (rRMSE) as evaluation metrics [18]. Then, the effectiveness of the calibration in the FPSF-CH was compared to the sub-model results without calibration (i.e., C-CH vs. RF-CH). The effectiveness of the FPSF-CH among different land surface characteristics as determined by different canopy cover and topographic slopes was then individually analyzed. Notably, a good model should be stable for canopy height mapping under different parameter settings. Thus, the robustness of the FPSF-CH was analyzed by comparisons of RMSE using different W values for weighting GEDI data and using a different number of GEDI observations for calibration. The different scenarios of W included only using $W_{os}$, and only using $W_{sen}$, and then using both $W_{os}$ and $W_{sen}$. The scenarios for different numbers of GEDI observations for calibration included the use of 5, 10, 15, 20, 25, and 30 samples.

## 3. Results

*3.1. Accuracies of Canopy Height Maps Compared with GEDI Validation Data*

The performance of canopy height mapping derived from the FPSF-CH (C-CH) was compared with high-quality GEDI validation data (Table 3 and Figure 3). The accuracies of the C-CH were also compared with the GDAL RH95. The results show that the C-CH overestimates canopy heights in Area 1, but underestimates canopy height in Area 2 with biases of 0.61 and −0.20 m, respectively. The C-CH shows more favorable accuracies than the accuracies of GDAL RH95 with a Pearson r improvement of 0.03 and 0.12, and reductions in absolute bias, MAE, RMSE, absolute rBias, and rRMSE of 0.67 and 2.52 m, 1.11 and 1.30 m, 1.64 and 1.95 m, 9.47 and 10.86%, and 8.19 and 8.41% in Area 1 and Area 2, respectively. The canopy height displayed in Figure 4 demonstrates the good performance of the FPSF-CH in distinguishing short from tall canopy heights, with a darker green area representing tall canopy heights, while light green represents short canopy heights (Figure 4(C1,C3)). Compared with the GDAL RH95 (Figure 4(A4)), C-CH under-expresses short canopy heights in Area 2 (Figure 4(A3)). This under expression might explain the low correlation (Pearson r) between C-CH and GEDI RH95, which is partly caused by (1) satellite imagery that is insensitive to canopy height, and (2) the coherence that exists among various forest stands with mixed trees with different stand age, species, crown shape, etc. Moreover, the topographic slope has less effect on the canopy height map of both GDAL RH95 and C-CH over space (Figure 4(A2–A4)), which demonstrates less topographic slope effects on land surface reflectance and the conservative choice of using slope values under 15 degrees for selecting high-quality GEDI for model training.

**Table 3.** Accuracies of the FPSF-CH for canopy height mapping (C-CH) and GDAL RH95 compared to high-quality GEDI validation data (GEDI RH95). The negative bias represents an underestimation compared to GEDI RH95. The asterisk represents a significant correlation ($p < 0.05$).

|  | Study Area | r | Bias (m) | MAE (m) | RMSE (m) | rBias (%) | rRMSE (%) |
|---|---|---|---|---|---|---|---|
| C-CH | Area 1 | 0.58 * | 0.61 | 3.48 | 4.47 | 3.06 | 22.34 |
|  | Area 2 | 0.36 * | −0.20 | 2.64 | 3.22 | −0.88 | 13.91 |
| GDAL RH95 | Area 1 | 0.55 * | −1.28 | 4.60 | 6.11 | −6.41 | 30.53 |
|  | Area 2 | 0.24 * | −2.72 | 3.94 | 5.17 | −11.74 | 22.32 |

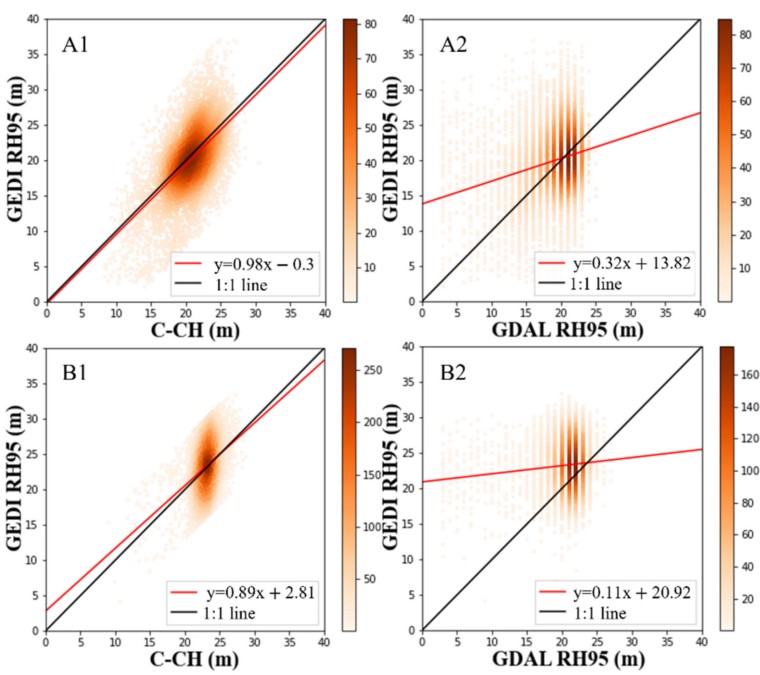

**Figure 3.** The relationship between canopy heights of C-CH (**A1**,**B1**) and GDAL RH95 (**A2**,**B2**) compared to high-quality GEDI validation data in Area 1 (**A**) and Area 2 (**B**), respectively.

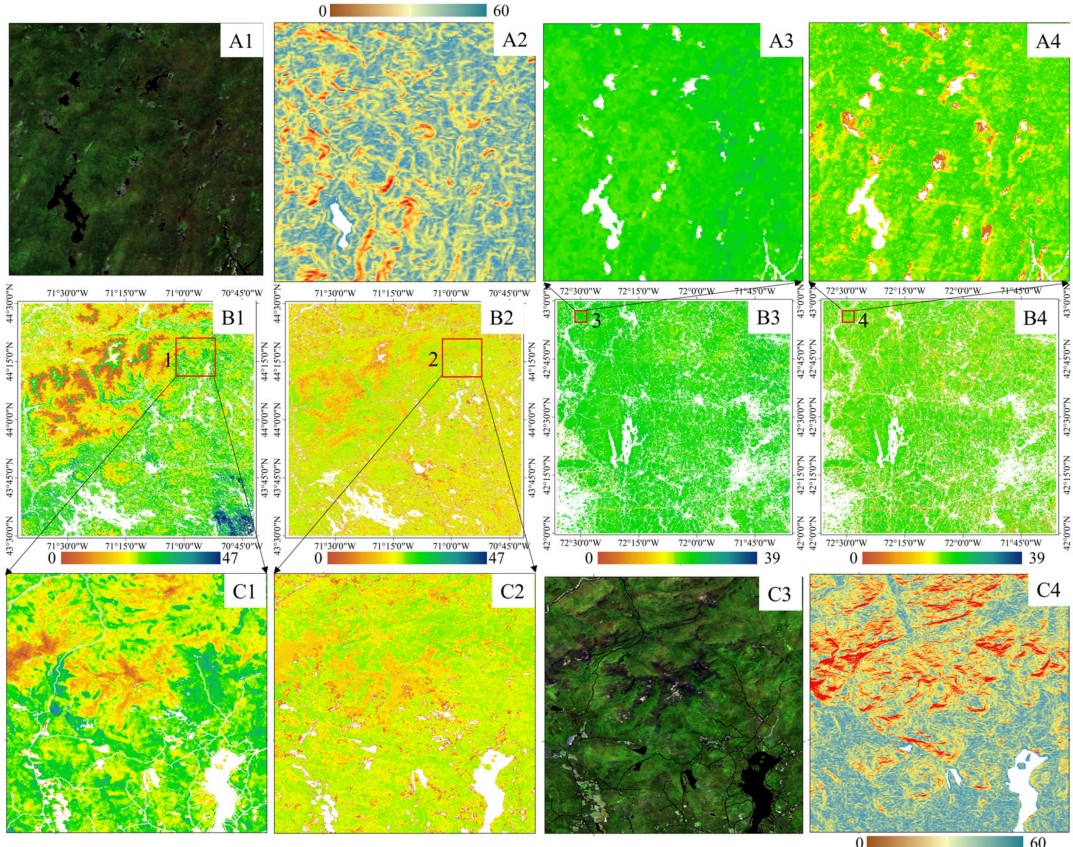

**Figure 4.** Comparison of canopy height spatial distributions in Area 1 (**B1**,**B2**) and Area 2 (**B3**,**B4**). (**A1**,**C3**) are true color images, while (**A2**,**C4**) are the topographic slopes of the enlarged rectangle in Area 1 (**C1**,**C2**) and Area 2 (**A3**,**A4**), respectively. (**B1**,**B2**), and (**B3**,**B4**) are C-CH and GDAL RH95 in Area 1 and Area 2, respectively. The true color images are composited with the blue, green, and red bands of Sentinel-2. Masked areas are displayed in white.

### 3.2. Accuracies of Canopy Height Maps Compared with NEON RH95

Compared with NEON RH95 (Table 4), the accuracies of C-CH show prior to GDAL RH95 with absolute bias reductions of 0.91, 2.83, and 4.08 m, MAE reductions of 0.84, 1.85, and 1.85 m, RMSE reductions of 1.28, 2.25, and 2.23 m, rBais reductions of 4.44, 11.34, and 16.22%, rRMSE reductions of 6.29, 9.01, and 8.9% in BART, HARV, and HOPB, respectively. The effectiveness of the FPSF-CH for canopy height mapping is also demonstrated by its increasing correspondence with NEON RH95 (Figure 5). Moreover, canopy height spatial distributions show that both the C-CH and GDAL RH95 reflect the full range (high and low) of canopy heights of NEON RH95 (Figures 5(A1–A3) and 6(A2)). Notably, the correlations between C-CH and NEON RH95 are site-specific, with the highest value of 0.58 in BART and the lowest value of 0.16 in HOPB. Variation in correlation strengths among different sites can be explained by the low to medium explanatory ability of wall-to-wall predictors for RF-CH prediction (Appendix A Figure A1). For example, a low correlation between canopy height and wall-to-wall imagery corresponds to the low r values of the FPSF-CH against NEON RH95 in HARV and HOPB, and vice versa in BART. Moreover, the site-specific correlations are probably caused by the explanatory capability of different resolution imagery for a heterogeneous landscape (Figure 6; Appendix A Figure A2). Figure 6(C2,C3) show a more obvious over-expression of short canopy heights in the GDAL RH95 rather than in the area in the C-CH in a heterogeneous landscape.

**Table 4.** Accuracies of the FPSF-CH for canopy height mapping (C-CH) and GDAL RH95 compared with NEON RH95. The negative bias represents an under-estimation compared with NEON RH95. The asterisk represents a significant correlation ($p < 0.05$).

| Metrics | C-CH | | | GDAL RH95 | | |
|---|---|---|---|---|---|---|
| | **BART** | **HARV** | **HOPB** | **BART** | **HARV** | **HOPB** |
| r | 0.58 * | 0.26 * | 0.16 * | 0.43 * | 0.22 * | 0.22 * |
| Bias (m) | −0.84 | −1.62 | −0.04 | −1.75 | −4.45 | −4.12 |
| MAE (m) | 2.96 | 3.16 | 2.66 | 3.8 | 5.01 | 4.51 |
| RMSE (m) | 3.82 | 4.05 | 3.48 | 5.1 | 6.3 | 5.71 |
| rBias (%) | −4.15 | −6.49 | −0.16 | −8.59 | −17.83 | −16.38 |
| rRMSE (%) | 18.77 | 16.24 | 13.81 | 25.06 | 25.25 | 22.71 |

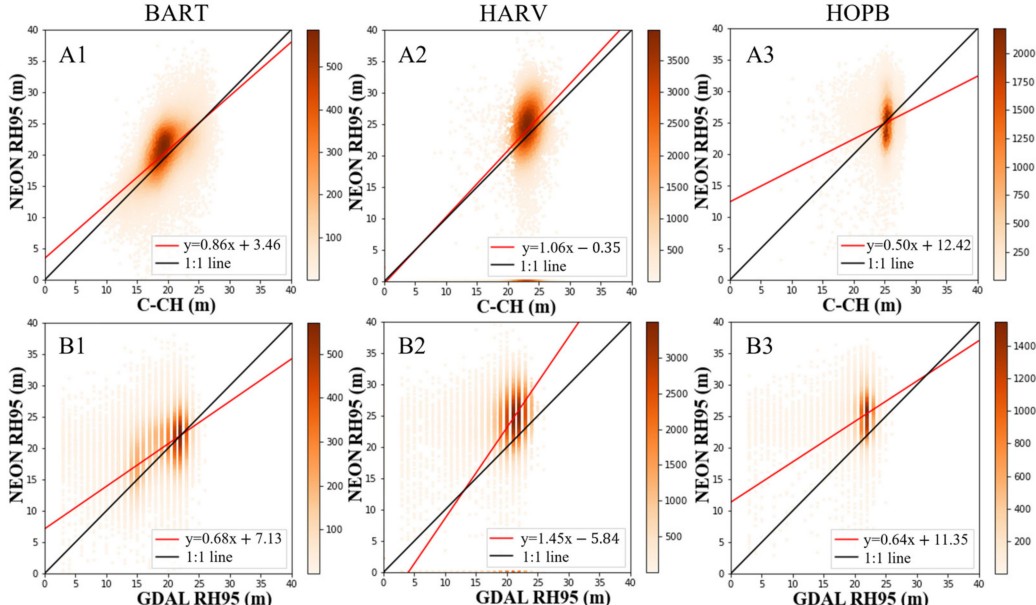

**Figure 5.** The relationship of canopy heights of C-CH (**A1–A3**) and GDAL RH95 (**B1–B3**) compared with NEON RH95 in BART (**A1,B1**), HARV (**A2,B2**), and HOPB (**A3,B3**), respectively.

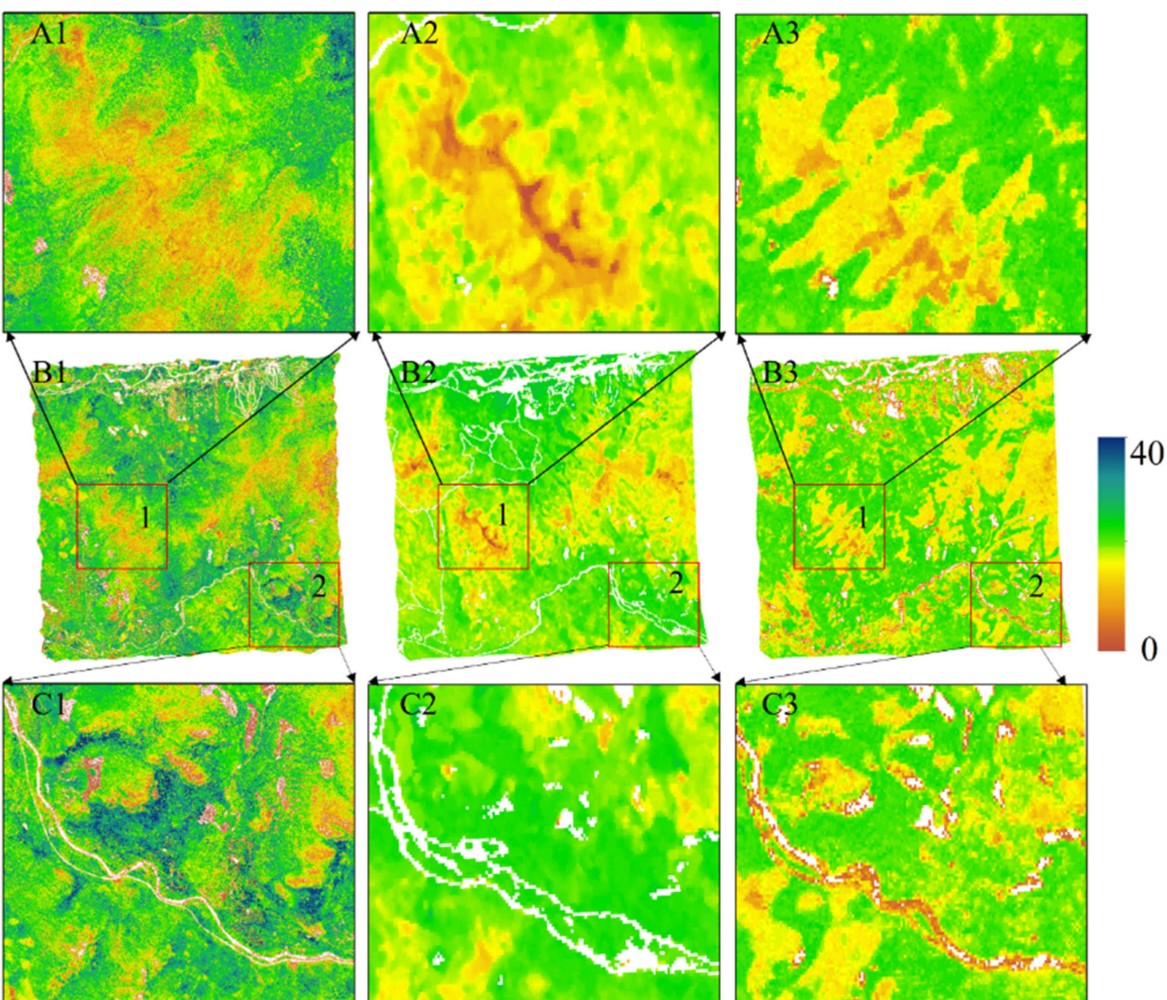

**Figure 6.** Comparison of canopy height spatial distributions among NEON CHM, C-CH, and GDAL RH95. The (**A1**–**A3**,**C1**–**C3**) are the sub-regions of '1', and '2' rectangles in (**B1**–**B3**). The (**B1**–**B3**) represent the NEON CHM, C-CH, and GDAL RH95, respectively. To be comparative, the canopy heights have the same scale, ranging from 0 to 40 m. Masked areas are displayed in white.

### 3.3. Accuracies of Canopy Height among Different Land Surface Characteristics

The canopy height residuals of C-CH derived from the FPSF-CH were quantified by comparisons with NEON RH95, which varied with slope and tree canopy cover. As displayed in Figure 7, the slope and tree canopy cover have a slight effect on C-CH accuracy, with a bias near zero (Figure 7(A1–A3,B1–B3)). Tree canopy covers exhibit greater variation, especially for those lower than 50% compared to the slope factor (Figure 7(B1–B3,D1–D3) correspondingly compared with Figure 7(A1–A3,C1–C3)). The greater residuals also appear in GDAL RH95 in areas with canopy covers lower than 50% (Figure 7(D1–D3)). Compared with a greater than 4 m underestimation of GDAL, the FPSF-CH alleviates the underestimation of canopy heights by integrating good-quality GEDI data. The underestimation of canopy height in low canopy cover areas is due partly to the high heterogeneity in forest stands, in which young forest stands are intermixed with mature forest and (or) soil (termed a mixed pixel), resulting in an underestimation of canopy height.

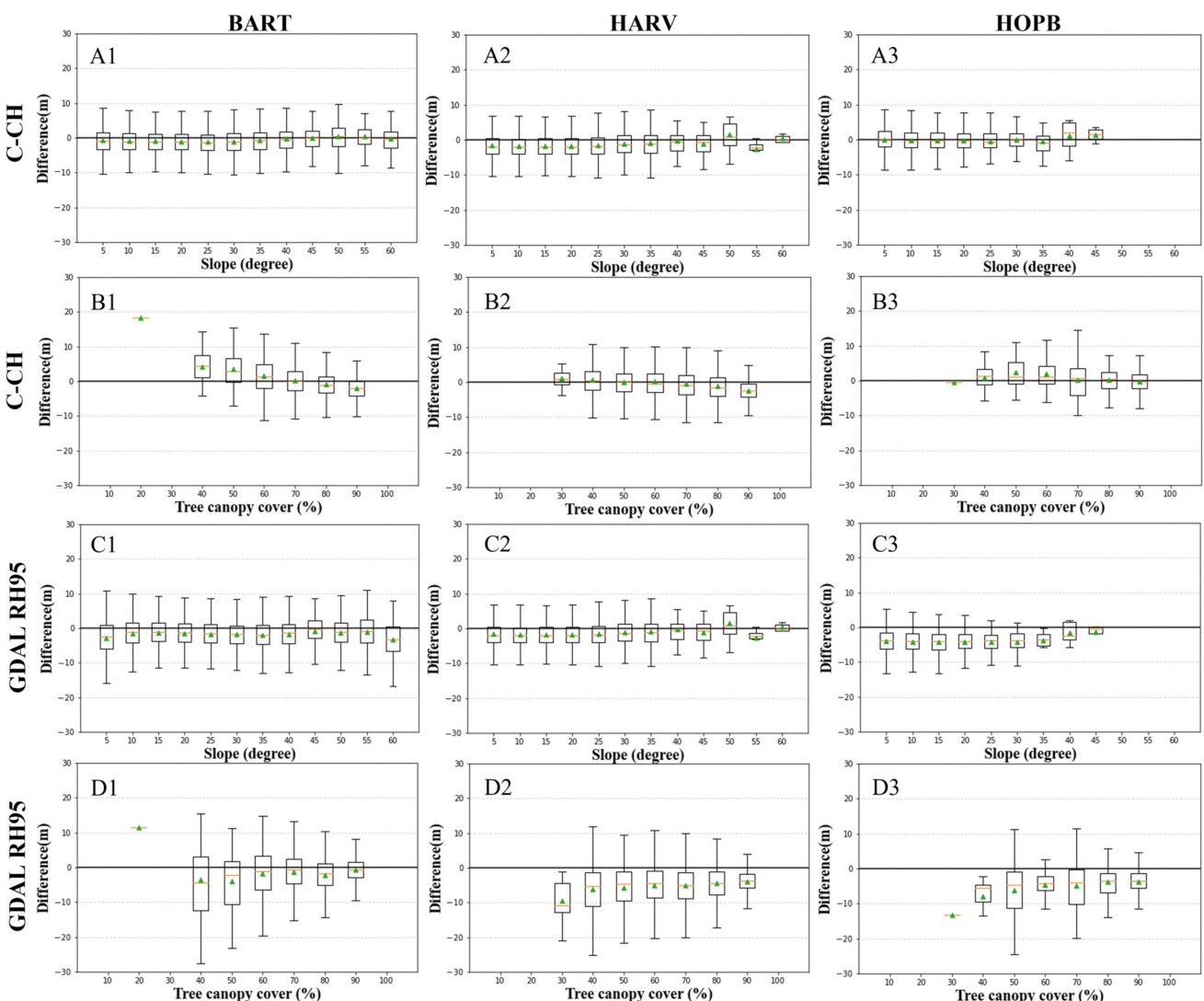

**Figure 7.** Canopy height residuals of C-CH (**A**,**B**) and GDAL RH95 (**C**,**D**) plotted as a function of slope (**A1**–**A3**,**C1**–**C3**) and tree canopy cover (**B1**–**B3**,**D1**–**D3**) in BART, HARV, and HOPB. To be readable, the median and average of residuals are shown by the orange line and a green triangle in the box, respectively. The negative median and average values indicate under-and over-estimation, respectively. The upper and lower quartiles are indicated by the end of the lines.

### 3.4. Accuracy Variations of Different Parameter Settings in the FPSF-CH

In essence, the FPSF-CH modifies the RF-CH by integrating a calibration procedure. Thus, the effectiveness of the calibration can be calculated by an accuracy comparison between C-CH and RF-CH (Tables 3–5). The comparisons indicate that the calibration enhanced the canopy height mapping by reductions in the absolute bias of 0.90, 0.44, and 2.02 m, reductions in MAE of 0.84, 0.34, and 0.84 m, reductions in RMSE of 1.28, 0.49, and 1.06 m, reductions in absolute rBias of 4.44, 1.76, and 8.10%, and reductions in rRMSE of 6.29, 1.95, and 4.38% in BART, HARV, and HOPB, respectively. The accuracy improvements of C-CH compared with RF-CH confirm the promising potential of integrating good-quality GEDI data for canopy height mapping. The comparison of canopy height between RF-CH and C-CH over space across Area 1 and Area 2 shows the superiority of the FPSF-CH for canopy height mapping (Figure 8(B1) vs. Figure 8(B2), Figure 8(B3) vs. Figure 8(B4)). As indicated by the enlarged rectangles 2 and 3 in Figure 8(C2,C3, the abnormal tall canopy height in RF-CH, especially for steep areas (enlarged rectangle 1 in Figure 8(C1) and fringe areas of different landscapes (enlarged rectangle 4 in Figure 8(C4)) can be corrected by

the FPSF-CH. For example, the abnormally tall canopy height in the bare rocky mountain cap in RF-CH (Figure 8(B2,C2)) is calibrated by FPSF-CH (Figure 8(B1,C1)). Another example shows that the abnormally tall canopy heights around the fringe areas in RF-CH (Figure 8(B4)) are calibrated into the normal canopy height of the focal areas (Figure 8(B3)). As a result of calibration, the results show a clearer distinction between short and tall vegetation amounts (red ellipse 1 in Figure 8(D1)) and a smoother distribution over tall canopy height areas (red ellipse 3 in Figure 8(D3)) in the C-CH histogram, correspondingly, compared with the RF-CH histogram in red ellipse 2 and 4 in Figure 8(D2), Figure 8(D4), respectively. Both the priority of the C-CH accuracy (Tables 3 and 4) and spatially correct canopy height distributions (Figure 8) reveal that calibration based on GEDI LiDAR observations can be a good method for precise canopy height mapping.

**Table 5.** Accuracy of RF-CH over the two study areas and three NEON sites compared with high-quality GEDI RH95 and NEON RH95, respectively. The asterisk represents a significant correlation ($p < 0.05$).

| Datasets | Sites | r | Bias (m) | MAE (m) | RMSE (m) | rBias (%) | rRMSE (%) |
|---|---|---|---|---|---|---|---|
| GEDI RH95 | Area 1 | 0.55 * | 0.91 | 3.50 | 4.66 | 4.57 | 23.26 |
| | Area 2 | 0.24 * | −0.39 | 2.93 | 3.73 | −1.69 | 16.10 |
| NEON RH95 | BART | 0.54 * | −0.29 | 3.33 | 4.46 | −1.43 | 21.9 |
| | HARV | 0.29 * | −2.06 | 3.5 | 4.54 | −8.25 | 18.19 |
| | HOPB | 0.22 * | −3.01 | 3.8 | 4.79 | −11.95 | 19.05 |

The FPSF-CH initially used RF to fuse GEDI observations with ancillary wall-to-wall imagery to estimate the continuous canopy height of forested areas. The effectiveness of the calibration process for canopy height mapping is influenced by the accuracy and availability of good-quality GEDI observations and other parameters. Figure 9 indicates a large amount of variability of good-quality GEDI observations over space within a 6 km width of searching windows, with values varying from 0 to 715 samples in Area 1, and from 0 to 946 in Area 2. The uniform distribution of observations in the form of strips over space ensures relevant stability of availability of GEDI observations for calibration, which avoids strips over calibrated maps (Figure 8(C1–C4)). Having less than 11% and 4% unfilled areas (Figure 9(A1,A2)) representing no GEDI observations in Area 1 and Area 2, respectively, further permits the possibility of using GEDI observations for calibrating canopy height maps over large-scale areas. Apart from their increased abundance, the better performance of good-quality GEDI data in canopy height measurements compared with the RF-CH further confirms the potential of using them in the calibration process (Table 6).

For the calibration process, two key steps were utilized, including one step for screening GEDI observations and another one for weighting each GEDI observation. In the screening step, a threshold of at least ten GEDI observations was used to determine if the corresponding RF-CH object pixel should be calibrated. To analyze the effect of this threshold on the performance of the FPSF-CH, comparisons were conducted among C-CH RMSE values using different thresholds ranging from 5 to 40. Figure 10A shows that the effect of the threshold values on RMSE values of C-CH varied among sites. Engaging more GEDI observations for the calibration process had a limited contribution to the improvement of the FPSF-CH, as indicated by an RMSE value reduction lower than 0.1 m in the BART and HARV (Figure 10A). For the HOPB site, the sets of five or ten GEDI observations were optimal for the FPSF-CH which hindered the RMSE values from increasing. A small number of observations was better than a large number of GEDI observations in HOPB for FPSF-CH performance, revealing that the FPSF-CH performance was influenced by the available GEDI observations over space. Using a small number of good-quality GEDI observations for calibration enabled broader areas to meet the requirements of calibration, and vice versa. Moreover, selecting a small number of representative observations using $W_{os}$ is good enough for FPSF-CH compared with using $W_{sen}$ and using both $W_{os}$ and

$W_{sen}$ which contributes to slight reductions in RMSE values compared with just using $W_{os}$ (Figure 10B).

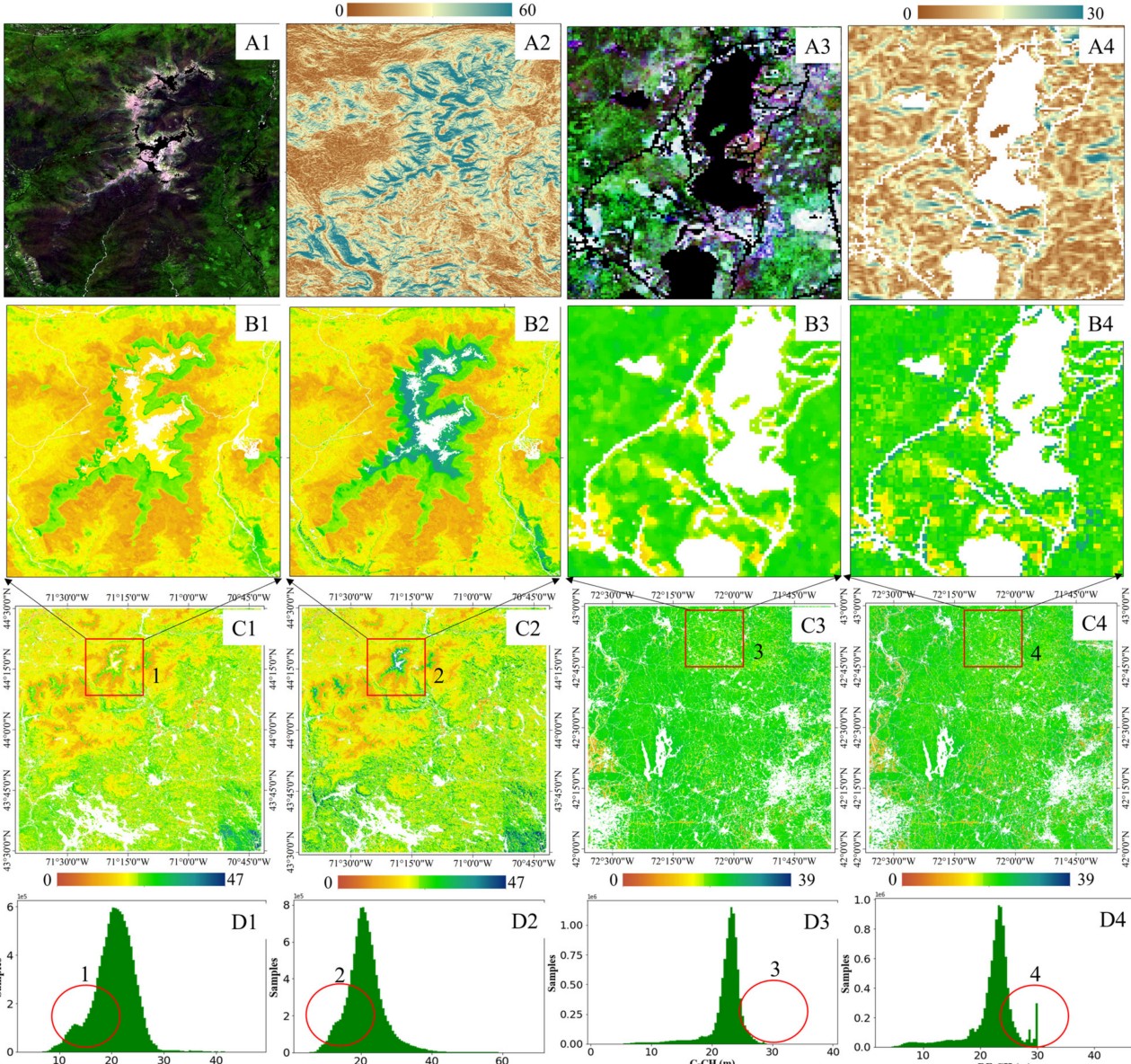

**Figure 8.** Comparison of distributions between C-CH (**C1**,**C3**) and RF-CH (**C2**,**C4**) over space and height histograms. (**A1**,**A3**) are true color images, while (**A2**,**A4**) are topographic slopes for the enlarged rectangle in Area 1 (**B1**,**B2**) and Area 2 (**B3**,**B4**). (**D1**–**D4**) represents histograms of canopy height of C-CH (**D1**,**D3**) and RF-CH (**D2**,**D4**) in Area 1 (**D1**,**D2**) and Area 2 (**D3**,**D4**), respectively. The red circles in (**D1**–**D4**) show the obvious difference between the canopy heights of RF-CH and C-CH. The true color images are composited with the blue, green, and red bands of Sentinel-2. Masked areas are displayed in white.

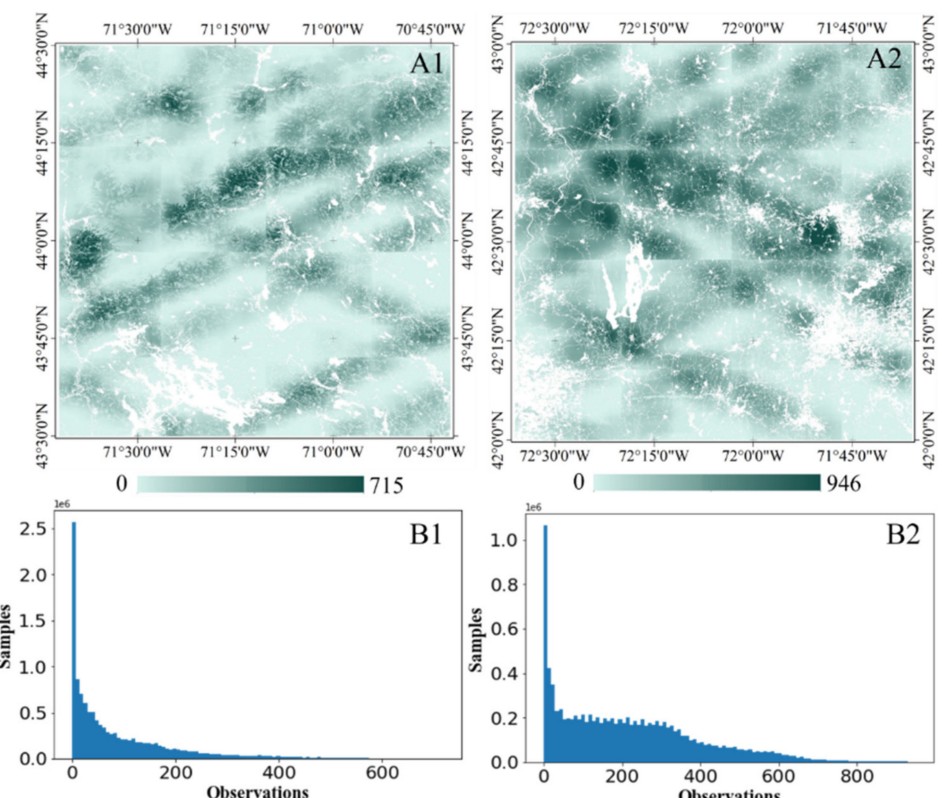

**Figure 9.** Densities of GEDI observations over space (**A1**,**A2**) and amounts (**B1**,**B2**) within a 6 km width window for each object pixel in RF-CH in Area 1 and Area 2, respectively. Masked areas in (**A1**,**A2**) are displayed in white.

**Table 6.** Accuracy comparison of canopy heights between good-quality GEDI data and RF-CH, compared with NEON RH95. r is the Pearson r correlation, while the asterisk represents a significant correlation ($p < 0.05$).

| Datasets | r | Bias (m) | MAE (m) | RMSE (m) |
|---|---|---|---|---|
| Good-quality GEDI data | 0.81 * | −2.11 | 2.63 | 3.55 |
| RF-CH | 0.54 * | −1.69 | 3.71 | 4.77 |

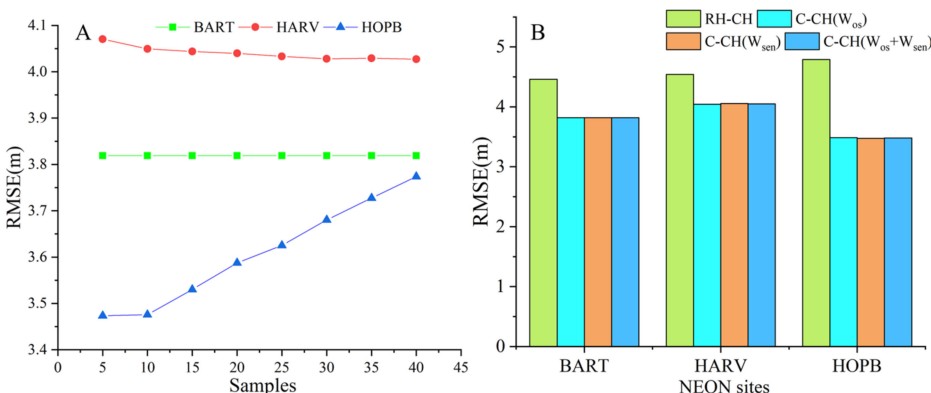

**Figure 10.** Performance of the FPSF-CH using a different number of good-quality GEDI observations (**A**) and weighting strategies (**B**) for calibration. The $W_{os}$ and $W_{sen}$ represent weights indicated by the DI difference between an object pixel in RF-CH and the corresponding GEDI data and waveform sensitivity, respectively.

## 4. Discussion

### 4.1. Adaptability of Intergrating GEDI-LiDAR in the FPSF-CH

GEDI is a next-generation spaceborne LiDAR that can revolutionize global measurements of vertical vegetation structures with its densely revisited observations. GEDI data is spatially discontinuous. As a result, GEDI should be combined with ancillary remote sensing datasets for wall-to-wall canopy height mapping. In this way, high-quality GEDI is generally employed as field-measured true canopy height values, resulting in the underutilization of lots of good-quality GEDI data. For example, Potapov et al. only used power beam data with a sensitivity greater than 0.9, and elevation differences between the six GEDI waveform preprocessing algorithms were less than 2 m during nighttime, leaf-on conditions for low-elevation (<1500 m) and flat areas (topographic slope < 6°) [19]. Rishmawi et al. used the quality_flag equal to 1, degrade flag not equal to 1, and sensitivity greater than 0.95 to select high-quality GEDI data, which consequently removed the GEDI coverage beam representing half of the GEDI observations [20]. As a result of data filtering, at least 50% of the GEDI good-quality observations that can possess substantial information content were unutilized [18]. Conversely, the proposed FPSF-CH, which integrates all available good-quality GEDI data with the antecedent canopy height map, can take full advantage of all good-quality GEDI observations for seeking improvements in accurate wall-to-wall canopy height mapping.

The primary novelty of the proposed FPSF-CH method is the calibration process, which uses a weighted regression to build the relationship between GEDI observations and each object pixel in the RH-CH. The employment of a weighted regression model comprehensively considered the difference in land cover type and the spectral difference between GEDI observations and a target pixel in RF-CH. By doing so, the FPSF-CH can effectively avoid errors caused by unreasonable observations against a target pixel in the RF-CH. Benefiting from this weighted regression model, the FPSF-CH has the potential to increase the accuracy of the canopy height map in heterogeneous areas. Its adaptability can be demonstrated by the better performance of the FPSF-CH compared with the sub-model of the canopy height, which is without integrating the GEDI LiDAR [16,21,23].

The validity of the calibration using GEDI LiDAR can be attributed to the following factors: (1) directly measuring the canopy height by GEDI data, which outperforms RF-CH in the canopy height estimations due to the beneficial effect of LiDAR penetration on canopy height measurements (Table 6). (2) Compared with high-quality GEDI observations, good-quality GEDI observations have a considerable number of independent shots (Table 2) at a comparable accuracy for canopy height measurements. The comparability of good-quality GEDI observations against high-quality GEDI data can be demonstrated by (1) comparing the measuring ability of canopy heights between these two datasets (Table 6), and (2) at least 15 times larger GEDI observations than that of high-quality GEDI data (Table 2). Thus, this paper advocates for the use of good-quality GEDI observations to correct the wall-to-wall canopy height maps based on predictors of wall-to-wall imagery.

### 4.2. Potentials of the FPSF-CH for Large-Scale Canopy Height Mapping

Precise large area canopy height mapping relies on (1) an accurate model, (2) data support, and (3) the efficiency of a model. The effectiveness of the FPSF-CH in this paper is confirmed by accuracy comparisons between C-CH and GEDI high-quality data in two study areas of $1 \times 1°$ tile (Figures 3 and 4; Table 3), and high-resolution NEON LiDAR at three sites (Figures 5 and 6; Table 4). For data support, the FPSF-CH aims to integrate GEDI data for improving the wall-to-wall canopy height map generated from the integration of field-measured values with ancillary remote sensing data. Since ancillary remote sensing data includes more than 40 years of historical inventory datasets, the availability of GEDI data becomes a key factor for the application of the FPSF-CH. Luckily, a large amount of GEDI data can be collected over space and time. For example, Fayad et al. pointed out that more than 25 billion GEDI observations (approximately 90 TB) have been collected during its original 18 months [17]. GEDI observations have been produced during the

GEDI mission until at least September 2023. Liu et al. argued that the GEDI point cloud density can reach 400 points/km$^2$ in China, and the shot density is 120–800 times that of ICESat GLAS [49]. Further, Healey et al. demonstrated the possibility of searching for 100 neighborhoods in GEDI observations within a 3 km focal area [21]. The fact that there exists more than 15 times the total amount of high-quality GEDI data also provides convincing support for the proposed method. Regarding efficiency, this FPSF-CH adopts weighted linear regression for the calibration step. This regression method requires fewer samples than complex regression models, such as random forest (RF), multivariate adaptive regression splines, and lasso and elastic-net regularized generalized linear models [22]. As a result, the model has high operating efficiency. Considering it possesses high mapping accuracy, sufficient data support, and low method complexity, the FPSF-CH has a high potential for application to large area canopy height mapping.

*4.3. Limitations and Potential Improvements*

It is worth noting that modeling and input datasets together lead to unknown errors in the FPSF-CH in canopy height mapping. Firstly, the quality of RF-CH as an input to the calibration process directly affects the accuracy of C-CH (Appendix A Figure A3). The RF-CH was affected by the relationship between canopy height and wall-to-wall ancillary data directly. This paper employed a total of 49 ancillary images, which had a moderate to low correlation with canopy height (Appendix A Figure A1). The low correlation can potentially result in over-estimating short vegetation, but under-estimating tall vegetation [28]. Regardless of the under-estimation caused by spectral saturation, it is perhaps not surprising that spectral reflectance within a 30 m resolution pixel is from both over-story and under-story. Thus, spectral reflectance is not necessarily related to canopy height. As described in [24,26], the over-estimation occurs mostly in regenerating forest stands that have high spectral reflectance from an under-story mixed up with a thinned over-story and its shadow, resulting in misleading high coverage and tree density [3]. Thus, using spectral information to predict canopy height can be an ill-posed task because the spectral reflectance of short vegetation is indistinguishable from high vegetation [24]. Engaging time-series and high-resolution imagery to characterize tree age and spatial context, respectively, can be a solution to this over-estimation issue [35]. How the spectral variables from multiple remote sensing datasets can be used for better canopy height mapping should be further researched.

The second major concern is the limited amount of GEDI data in steep areas due to the fact that observations in areas with a topographic slope greater than 25 degrees should be discarded. The sparse GEDI data around steep areas sometimes causes no data for calibration. An auto-adaptive moving window strategy can be a way to expand the search region for GEDI data for calibration. An alternative way to expand GEDI data is to recover canopy height derived from GEDI waveforms over sloping terrain. Optional methods for precision canopy height correction in a steep area include (1) Lefsky's model, which corrects canopy height in a steep area using an experimental function between the response variable of the true canopy height and explanatory variables of large-size waveform extent [50], (2) Lee's model, which removes terrain effect using a physical relationship between inclined ground elevation and topographic slopes [51], and (3) Wang's model, which proposes using slope-adaptive waveform metrics to replace relative height [47,52]. In addition, using other spaceborne LiDAR datasets from ICESat ATLAS and the upcoming multi-footprint observation LiDAR and imager system (MOLI) is expected to further expand the GEDI data.

It should be noted that the FPSF-CH consists of two serially connected processes, in which the RF model was employed to generate the antecedent wall-to-wall canopy height maps for the calibration process. These two step-by-step processes will impact the efficiency of the FPSF-CH. To improve efficiency, future work may focus on developing a parallelized model for wall-to-wall canopy height mapping and calibration with GEDI data. Finally, weighting each corresponding GEDI observation is important for the FPSF-CH. Correction factors including waveform sensitivity and spectral difference have been shown

to be effective for RH-CH calibration (Figure 10), but the potential to introduce errors should be emphasized. The weighting processes could be confounded by misclassified land cover types being used to identify GEDI observations. Thus, potential improvements for weighting each corresponding GEDI observation can be discussed in a future study.

## 5. Conclusions

This paper proposes a framework for point-surface fusion for canopy height mapping (FPSF-CH). The FPSF-CH allows one to integrate spatially scattered good-quality GEDI observations with a spatially continuous antecedent canopy height map based on interpolation using optical/SAR imagery. In this paper, the effectiveness of the FPSF-CH was demonstrated with its better accuracy compared with the GDAL RH95 and the antecedent canopy height map without integration (RF-CH). Meanwhile, by employing weighted regression to weight GEDI observations for calibration, the FPSF-CH was validated to work well in a heterogeneous area. Furthermore, the FPSF-CH is validated as more robust for canopy height mapping with a lower and near-zero bias (compared with the GDAL-RH95) over areas with different slopes and canopy cover values. With low computational complexity (e.g., high efficiency) and near-global availability of GEDI data, the proposed FPSF-CH method has a high potential for large-scale or even global canopy height mapping.

**Author Contributions:** Conceptualization, C.W. and A.J.E.; Funding acquisition, C.W. and M.A.C.; Methodology, C.W. and A.J.E.; Resources, C.W., A.J.E., I.N., M.A.C., S.L. and C.R.H.; Software, C.W.; Supervision, A.J.E.; Validation, C.W. and C.R.H.; Writing—original draft, C.W.; Writing—review and editing, C.W., A.J.E., I.N., M.A.C., S.L., C.R.H., Y.Z., Y.L. and Y.T. All authors have read and agreed to the published version of the manuscript.

**Funding:** This work is supported by the joint Ph.D. program of "double first rate" construction disciplines of CUMT.

**Data Availability Statement:** The raw discrete point cloud LiDAR (NEON LiDAR) and GEDI data were generated at the National Ecological Observation Network (NEON) and the Land Processes Distributed Active Archive Center (LP DAAC), respectively. Derived data supporting the findings of this study are available from the corresponding author on request.

**Acknowledgments:** The participation of Mark A. Cochrane, Andrew J. Elmore, and Izaya Numata was supported by NASA LCLUC (80NSSC20K0365). The participation of Shaogang Lei was supported by the Ordos Science & Technology Plan (2021EEDSCXQDFZ010). We also would like to thank Duo Jia, who provided suggestions for revising the manuscript. Moreover, we would like to thank the four anonymous reviewers for providing us with constructive comments and suggestions, which helped us to improve the quality of this manuscript.

**Conflicts of Interest:** The authors declare no conflict of interest.

## Appendix A

**Table A1.** NEON LiDAR system for aerial laser scanning (ALS) platforms.

| System Parameters | Optech ALTM Gemini (ALS) |
|---|---|
| Altitude | 1 km |
| Returns per pulse | 4 |
| Wavelength | 1064 nm |
| Measurement range | 150–4000 m |
| Range accuracy (typical) | ±5–30 cm |
| Bean divergence angle | $0.25 \times 0.8$ mrad |
| Measurement rate (per second) | 0–70 (programmable) |
| Average point density | 1–4 points/m$^2$ |

**Table A2.** The features were extracted from Sentinel-1 and-2 and other datasets for modeling canopy height by RF. B1-12 corresponds to the band of Sentinel-1 datasets in which VV and VH are bands in Sentinel-2. Details of derivative features can be found in the index database (URL: https://www.indexdatabase.de/db/is.php?sensor_id=96).

| Name | Transformation | Data | Name | Transformation | Data |
|---|---|---|---|---|---|
| Spectral bands | B2,B3,B4,B5,B6,B7,B8,B8A,B11,B12 | | S2REP | $705 + 35 \times ((B4 + B7)/2 - B5)/(B6 - B5)$ | |
| CIg-re1 | $B5/B3 - 1$ | | SR | $B8/B4$ | Sentinel-2 |
| CIg-re2 | $B6/B3 - 1$ | | TNDVI | $\sqrt{((B8 - B4)/(B8 + B4) + 0.5)}$ | |
| CIg-re3 | $B7/B3 - 1$ | | SCLAST | Time since last change ranging from 1982 to 2017 | USGS LCMAP |
| CIg | $B8/B3 - 1$ | | Elevation | Elevation | SRTM |
| DVI | $B8 - B4$ | | Slope | Slope | DEM |
| EVI | $2.5 \times (B8 - B4)/(B8 + B4 \times 6 - 7.5 \times B2 + 1)$ | Sentinel-2 | G-RVI | Growing season radar vegetation index | |
| FDI | $B8 - (B3 + B4)$ | | NG-RVI | Non-growing season radar vegetation index | |
| IRECI | $(B7 - B4)/(B5/B6)$ | | G-VVmean | Growing season means VV | |
| MCARI | $((B5 - B4) - 0.2 \times (B5 - B3)) \times (B5/B4)$ | | G-VVmedian | Growing season median VV | |
| MDI1 | $(B8 - B11)/B11$ | | G-VVstd | Growing season VV standard deviation | Sentinel-1 |
| MDI2 | $(B8 - B12)/B12$ | | NG-VVmean | Non-growing season means VV | |
| MNDWI | $(B3 - B11)/(B3 + B11)$ | | NG-VVmedian | Non-growing season median VV | |
| MSRren | $\sqrt{((B8A/B5) - 1)/((B8A/B5) + 1)}$ | | NG-VVstd | Non-growing season VV standard deviation | |
| MTCI | $(B6 - B5)/(B5 - B4)$ | | G-VHmean | Growing season mean VH | |
| NDVI | $(B8 - B4)/(B8 + B4)$ | | G-VHmedian | Growing season median VH | |
| NDVIre1 | $(B8 - B5)/(B8 + B5)$ | | G-VHstd | Growing season VH standard deviation | |
| NDVIre2 | $(B8 - B6)/(B8 + B6)$ | | NG-VHmean | Non-growing season mean VH | |
| NDVIre3 | $(B8 - B7)/(B8 + B7)$ | | NG-VHmedian | Non-growing season median VH | |
| PSSRa | $B7/B4$ | | NG-VHstd | Non-growing season VH standard deviation | |

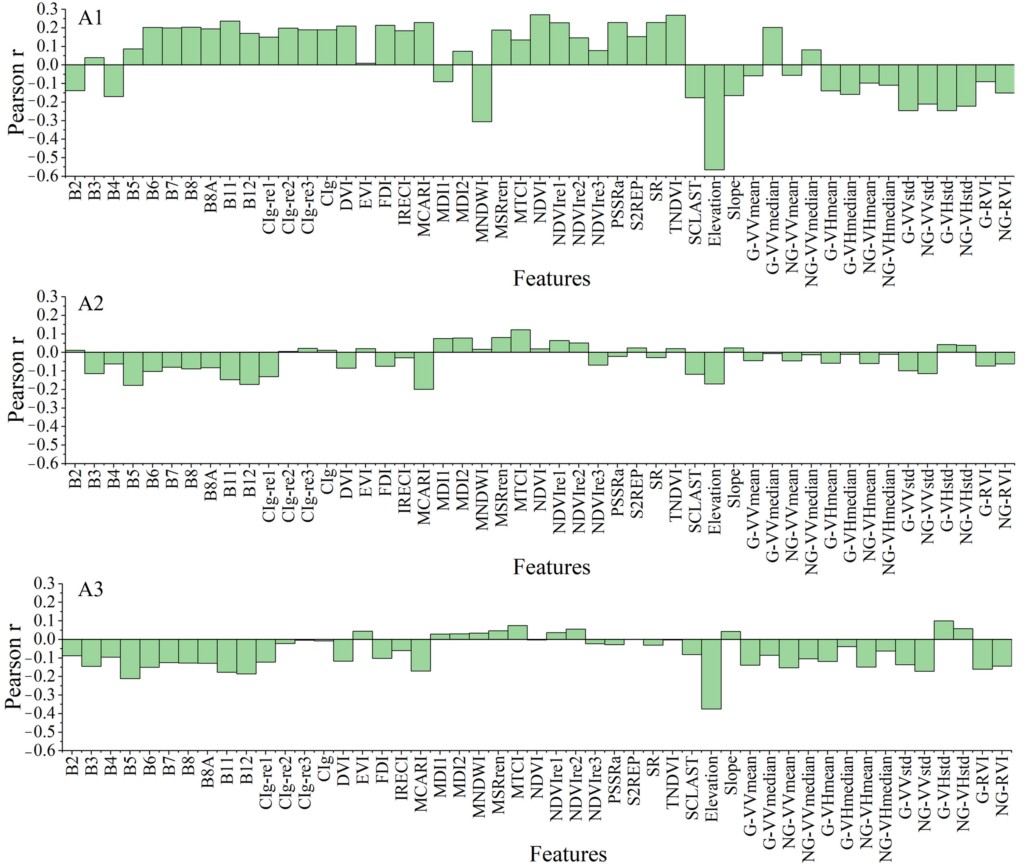

**Figure A1.** Correlations between canopy height and features derived from ancillary images in BART (**A1**), HARV (**A2**), and HOPB (**A3**). Explanations of each feature can be found in Table 4.

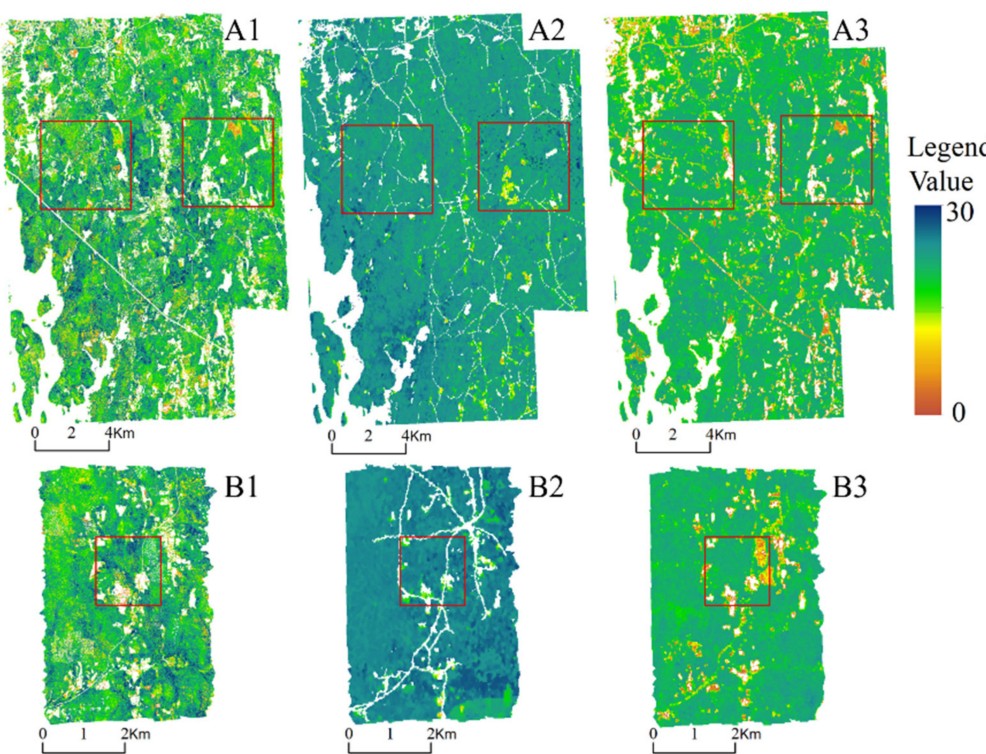

**Figure A2.** The distribution of NEON RH95 (**A1**,**B1**), C-CH (**A2**,**B2**), and GDAL height (**A3**,**B3**) in HARV (**A**) and HOPB (**B**) sites.

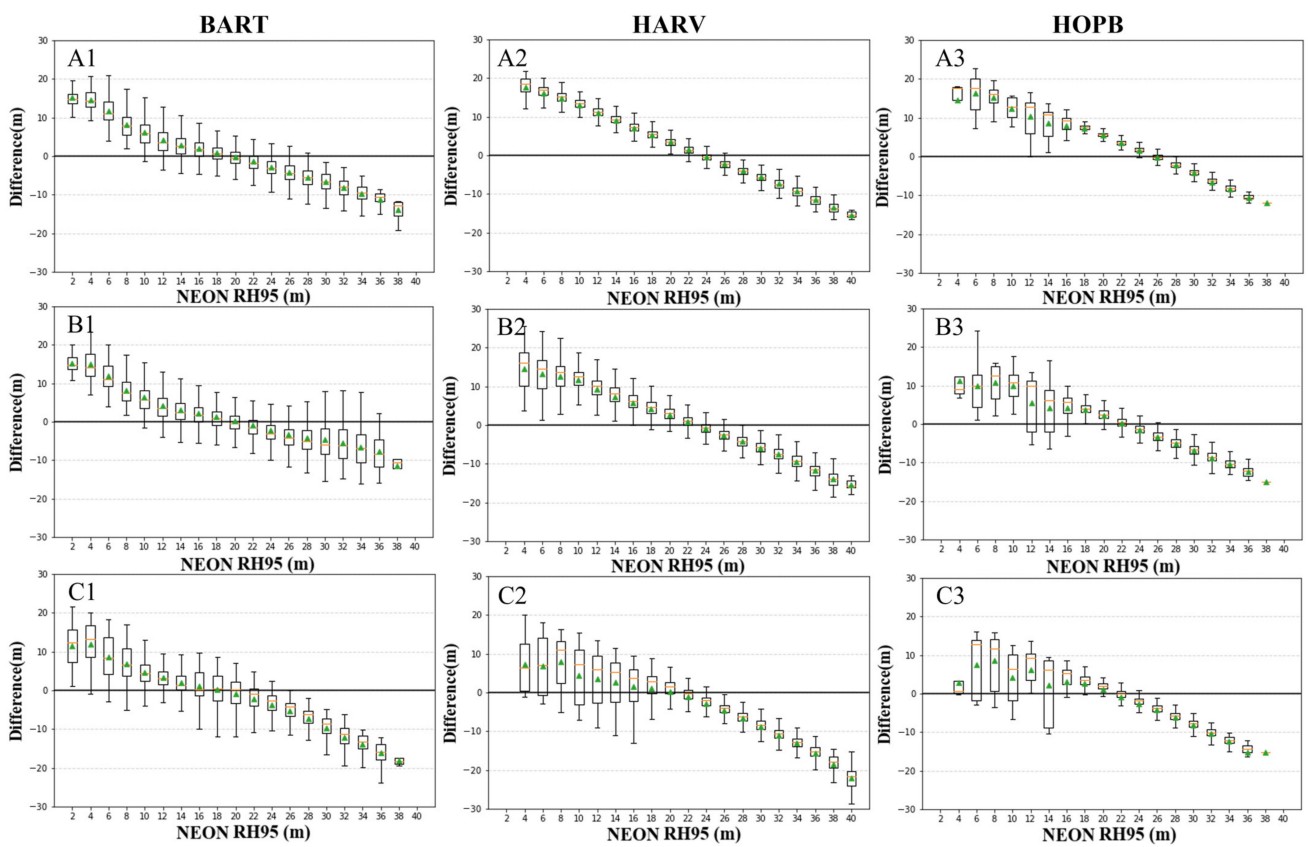

**Figure A3.** The residuals of the C-CH (**A1**–**A3**), RF-CH (**B1**–**B3**), and GDAL RH95 (**C1**–**C3**) among different canopy heights measured with NEON LiDAR in BART, HARV, HOPB.

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
