# Peer review of "A Framework for Improving Wall-to-Wall Canopy Height Mapping by Integrating GEDI LiDAR"

_remotesensing, doi:10.3390/rs14153618_

Round 1

Reviewer 1 Report

This point-surface fusion method for canopy height mapping is novel and should make a positive contribution to the literature.  I have no major concerns or suggestions/revisions for this manuscript and feel confident that it will make a positive impact.  It is great to see GEDI data being used and the continuation of the mission through 2023.

Author Response

Comments from the Editors and Reviewers:

Note: line numbers used in response part below refer to the untracked version of the revised manuscript.

Reviewer 1

This point-surface fusion method for canopy height mapping is novel and should make a positive contribution to the literature. I have no major concerns or suggestions/revisions for this manuscript and feel confident that it will make a positive impact. It is great to see GEDI data being used and the continuation of the mission through 2023.

Response: We appreciate for your affirmation about our manuscript. Thanks for your valuable comments. Yes, it is so great that GEDI will collect waveform lidar until 2023. We believe that GEDI as a new generation spaceborne LiDAR can revolutionize global measurements of vertical vegetation structure with its dense revisited observations.

Reviewer 2 Report

Canopy height is a critical parameter for the estimation and monitoring of above ground biomass. This research makes a framework for wall-to-wall canopy height mapping by integrating auxiliary GEDI Lidar. This is an interesting and intact research, but before it can be published in this journal. These points should be considered and improved.

1)     Abstract. The abstract should be rewrite because the scientific question is not clear. From line 17 to 25, the authors tried to explain the key issues they want to address, yet it not clears now. Results here are also confused and not clear.

2)     Keyword. Now, there are just three key words, add more.

3)     Introduction. The introduction should be rewrite because the key scientific question is not clear. Meanwhile, there are too many “however” in this part, please be careful before use this important adversative.

4)     Discussion. The discussion should be rewrite; the first paragraph here is not proper, e.g., from line 635-643. Part of 4.1, it more like a result.

5)     Conclusion. The conclusion should be checked, robust and clear conclusions are scarce. In the line 816, this sentence is not proper here.

Author Response

Comments from the Editors and Reviewers:

Note: line numbers used in response part below refer to the untracked version of the revised manuscript.

Reviewer 2

Canopy height is a critical parameter for the estimation and monitoring of above ground biomass. This research makes a framework for wall-to-wall canopy height mapping by integrating auxiliary GEDI Lidar. This is an interesting and intact research, but before it can be published in this journal. These points should be considered and improved.

Response: We appreciate for your affirmation about our manuscript. Thanks for your valuable suggestions and comments. We have corrected our manuscript according to your comments. The corresponding revision can be found with tracked corrections in the updated manuscript.

1)Abstract. The abstract should be rewrite because the scientific question is not clear. From line 17 to 25, the authors tried to explain the key issues they want to address, yet it not clears now. Results here are also confused and not clear.

Response: Thanks for your valuable suggestions and comments which make an important contribution to the improvement of our manuscript. The abstract has been rewritten and the scientific question has been reorgnized (line 15 to 21). The corresponding revision can be found with tracked corrections in the updated manuscript. The rewritten contents are as follows:

Line 17-25 in abstract: The global ecosystem dynamics investigation (GEDI) waveform can penetrate a canopy to precisely find the ground and measure canopy height, but are spatially discontinuous over the earth surface. A common method to achieve wall-to-wall canopy height map is to integrate a set of field-measured canopy heights and spectral bands from optical and/or microwave remote sensing data as ancillary information. However, due partly to the saturation of spectral reflectance to canopy height, the product of this method may misrepresent canopy height. As a result, neither GEDI footprints nor interpolated map using the common method can accurately produce spatially-continuous canopy height map alone.

2) Keyword. Now, there are just three key words, add more.

Response: Thanks for your valuable suggestions. We have extended the keyword part to 5 words accordingly. The corresponding revision can be found with tracked corrections in the updated manuscript.

Keywords: Canopy height; FPSF-CH; Calibration; GEDI LiDAR; Forest

3) Introduction. The introduction should be rewrite because the key scientific question is not clear. Meanwhile, there are too many “however” in this part, please be careful before use this important adversative.

Response: Thank you for your guidance. We have clarified and rewritten the introduction section according to your comments. Specifically, we focused on the fact that ‘neither GEDI footprints nor interpolated map using the common method can accurately produce spatially-continuous canopy height map alone’. To address this issue, we proposed the FPSF-CH to integrate GEDI lidar with the antecedent wall-to-wall canopy height map for seeking an improvement. Moreover, the word of ‘however’ in introduction section is used cautiously by removed most of this word from our manuscript. The corresponding revision can be found with tracked corrections in the updated manuscript. The rewritten contents are as follows:

Line 101-103 in introduction: As mentioned above, using GEDI data or the common method alone is problematic for accurate wall-to-wall canopy height mapping, but the two may be complementary when used together. Therefore, this paper proposes a framework of point-surface fusion for spatially-continuous canopy height mapping (FPSF-CH).

4) Discussion. The discussion should be rewrite; the first paragraph here is not proper, e.g., from line 635-643. Part of 4.1, it more like a result.

Response: Thanks for your patience and guidance. The discussion section has been improved by (1) reorganizing the structure of this section into ‘4.1. Adaptability of integrating GEDI-LiDAR in the FPSF-CH, 4.2. Potentials of the FPSF-CH for large-scale canopy height mapping, and 4.3. Limitations and potential improvements’, (2) deleting the first paragraph, (3) moving the original Part of 4.1 to the result section but delete the figure 11 due to its limited information (line 501-511 in the result section).  The corresponding revision can be found with tracked corrections in the updated manuscript.

5) Conclusion. The conclusion should be checked, robust and clear conclusions are scarce. In the line 816, this sentence is not proper here.

Response: We have rewritten the conclusion section by focusing on the functioning, accuracy, application potentials, and limitation of the proposed FPSF-CH. Moreover, the sentence in line 816 has been removed according to your comment, which can be seen in the updated manuscript.

Conclusion section: This paper proposes a framework of point-surface fusion for canopy height mapping (FPSF-CH). The FPSF-CH allows one to integrate spatially scattered good-quality GEDI observations with a spatially continuous antecedent canopy height map based on interpolation using optical/SAR imagery. In this paper, the effectiveness of the FPSF-CH was demonstrated by its better accuracy compared with the GDAL RH95 and the antecedent canopy height map without integration (RF-CH). Meanwhile, by employing the weighted regression to weight GEDI observations for calibration, the FPSF-CH was validated to work well in a heterogeneous area. Furthermore, the FPSF-CH is validated as more robust for canopy height mapping with a lower and near to zero bias (compared with the GDAL-RH95) over areas with different slopes and canopy cover values. With low computational complexity (e.g., high efficiency) and near-global availability of GEDI data, the proposed FPSF-CH method has high potential for large-scale or even global canopy height mapping.

Reviewer 3 Report

 1 - More limitations and information of lidar in forestry is needed in the introduction part (45-54). Try to add some information about it. Some suggested papers which can give you useful information:  https://doi.org/10.3390/rs4040950, https://doi.org/10.1080/19475705.2021.1964617, https://doi.org/10.3390/f6124390.

2 – Improve the paper’s template. Sometimes the text and table captions are stuck together and it is difficult to read.

3 – I really appreciate the links for the data collection. Good job!

4 -  Could you explain better the “stricter screening conditions” statement at line 388?

5 – Improve a little the table’s quality sometimes are confusion the reader.

6 – Uniform the way you write lidar, use LiDAR instead.

That’s all!

Author Response

Comments from the Editors and Reviewers:

Note: line numbers used in response part below refer to the untracked version of the revised manuscript.

Reviewer 3

1 – More limitations and information of lidar in forestry is needed in the introduction part (45-54). Try to add some information about it. Some suggested papers which can give you useful information: https://doi.org/10.3390/rs4040950, https://doi.org/10.1080/19475705.2021.1964617, https://doi.org/10.3390/f6124390.

Response: Thanks for your patience and guidance. We have added descriptions of the recommended limitations and information of lidar in forestry. Moreover, the recommended paper really broadens my horizons, and have been added in our manuscript. The corresponding revision can be found with tracked corrections in the updated manuscript. The rewritten contents are as follows.

Line 55-63: At present, several LiDAR platform types have been used for canopy height measurement, including terrestrial (including static and mobile platforms), airborne (including manned and unmanned aerial vehicles), and spaceborne LiDAR. Airborne and terrestrial LiDARs have high resolution, and are accurate in canopy height measurement, but are limited by coverage and the expensive and time consuming in data collection [10–12]. Meanwhile, accuracy of canopy height measurement using different sources of airborne LiDAR is influenced by point density, forest stand density, and tree size [13]. As a result, canopy height measurement over large-scale extents is a challenging endeavor.

2 – Improve the paper’s template. Sometimes the text and table captions are stuck together and it is difficult to read.

Response: We really appreciate your comments here which make our manuscript more readable than the original version. The paper’s template has been improved according to the journal’s guideline. The corresponding revision can be found with tracked corrections in the updated manuscript.

3 – I really appreciate the links for the data collection. Good job!

Response: Thank you so much for your confirmation here which encourage us to make our manuscript clearer and easily to follow.

4 – Could you explain better the “stricter screening conditions” statement at line 388?

Response: Thanks for your comments here, which make our manuscript more readable. We have corrected it according to this comment which can be seen in the updated manuscript with tracked corrections. The revised sentences are as follows.

Line 313-318: The high-quality GEDI data was proposed to be theoretically precise for canopy height measurement, and was selected by using stricter screening conditions than the above filtered good-quality GEDI observations. Compared to conditions for good-quality filtering, the stricter conditions for high-quality GEDI data filtering should also meet the following criteria: being sampled by power beams rather than coverage beams, being collected with sensitivity greater than 0.98, and possessing a slope lower than 15 degrees.

5 – Improve a little the table’s quality sometimes are confusion the reader.

Response: Thanks for your comments here. To make our manuscript more readable, the table of ancillary information for NEON discrete point cloud LiDAR (original Table 1) and the table of complex features for antecedent wall-to-wall canopy height mapping (original table 2) have been moved to appendix A. Moreover, to make the tables more readable, we further clarified the table content in title. The corresponding revision can be found with tracked corrections in the updated manuscript.

6 – Uniform the way you write lidar, use LiDAR instead.

That’s all!

Response: Thanks for your comments here. We have uniformed the word lidar in our revised manuscript. The corresponding revision can be found with tracked corrections in the updated manuscript.

Reviewer 4 Report

This study proposes a point-surface fusion method for canopy height mapping with the integration of auxiliary GEDI Lidar data. The manuscript was well prepared and structured. It is nearly ready for publication. I just have some minor comments:

-          I would suggest to change the title: “A framework for improving wall-to-wall canopy height mapping by integrating auxiliary GEDI lidar”. Approaches for wall-to-wall canopy height mapping from GEDI and Sentinel/Landsat have been demonstrated by previous studies. The novelty of this work is the integration of auxiliary GEDI lidar to improve the results.

-          The manuscript is generally too long, so it is a bit hard to follow. Most the sections should be condensed.

-          Table 8 and Figure 11 should be moved to the Result section.

Author Response

Comments from the Editors and Reviewers:

Note: line numbers used in response part below refer to the untracked version of the revised manuscript.

Reviewer 4

This study proposes a point-surface fusion method for canopy height mapping with the integration of auxiliary GEDI Lidar data. The manuscript was well prepared and structured. It is nearly ready for publication. I just have some minor comments:

Response: We appreciate for your affirmation about our manuscript. Thanks for your valuable suggestions and comments. We have corrected our manuscript according to your comments. The corresponding revision can be found with tracked corrections in the updated manuscript.

-I would suggest to change the title: “A framework for improving wall-to-wall canopy height mapping by integrating auxiliary GEDI lidar”. Approaches for wall-to-wall canopy height mapping from GEDI and Sentinel/Landsat have been demonstrated by previous studies. The novelty of this work is the integration of auxiliary GEDI lidar to improve the results.

Response: Thanks for your guidance here. We have changed the title as ‘A framework for improving wall-to-wall canopy height mapping by integrating GEDI LiDAR’. Corresponding to the updated title, we further updated our manuscript to fit to this title. Specifically, we deleted the ‘auxiliary’ word in title because this paper focus on point-surface fusion method which in fact is suitable for all good-quality GEDI data rather than only for auxiliary GEDI data. The corresponding revision can be found with tracked corrections in the updated manuscript.

-The manuscript is generally too long, so it is a bit hard to follow. Most the sections should be condensed.

Response: Thanks for your comment here, which makes our manuscript more readable. We have condensed contents carefully (without sacrificing the key research results) throughout this manuscript with the reductions of pages and words from 31 to 26 pages, and 13603 to 10771 words, respectively. The corresponding revision can be found with tracked corrections in the updated manuscript.

-Table 8 and Figure 11 should be moved to the Result section.

Response: Thanks for your guidance here. We have moved table 8 to result section but delete the figure 11 from revised manuscript due to its limited information (line 501-511 in result section). The corresponding revision can be found with tracked corrections in the updated manuscript.

Round 2

Reviewer 2 Report

The authors have improved the manuscript based on my suggestions.